# Selection, Reflection and Self-Refinement: Revisit Reasoning Tasks via a Causal Lens

**Yunlong Deng** [1]   **Boyang Sun** [1]   **Yan Li** [1]   **Lingjing Kong** [2]

**Zeyu Tang** [2,3]   **Kun Zhang** [1,2]   **Guangyi Chen** [1,2]

## Abstract

Due to their inherent complexity, reasoning tasks have long been regarded as rigorous benchmarks for assessing the capabilities of machine learning models, especially large language models (LLMs). Although humans can solve these tasks with ease, existing models, even after extensive pre-training and post-training at scale, still fail to perform reasoning reliably. In this paper, we revisit reasoning tasks from a causal perspective, seeking to understand their behavior in latent space and to offer insights for addressing their challenges. Specifically, we cast reasoning tasks as a selection mechanism, in which high-level logical concepts function as selection operators on the given observations, such as, identifying the correct answer in a math problem or filling the appropriate entry in Sudoku. We emphasize two key properties of this formulation that shed light on the difficulty of reasoning tasks. First, the latent space exceeds the observation space in complexity, even when the correct answer is fully determined by the observed input. Second, the latent variables, corresponding to logical thought, are densely structured and exhibit strong dependencies. Building on this formulation, we introduce a framework, called $\textbf{SR}^2$, that incorporates the estimated latent variables as feedback into the selection mechanism, thereby facilitating the learning of dense dependencies among latent representations. The framework consists of three key modules: reflective representation learning, dependency self-refinement, and periodic intermediate alignment. Experimentally, we show that our approach yields significant gains in reasoning accuracy, for example, attaining over $10\%$ improvement in performance with $8\times$ fewer parameters on the Sudoku and Maze tasks over the recent advances. Code available at https://github.com/dengyl20/SR2.

## 1 Introduction

> "We do not learn from experience... we learn from reflecting on experience."
>
> — John Dewey, *How We Think*

Reasoning, the act of deriving conclusions and principles from information by logical inference, lies at the heart of human intelligence. From solving arithmetic problems to navigating mazes, humans perform complex reasoning with ease and reliability. In contrast, machine learning models continue to struggle with these tasks. Reasoning challenges have therefore become one of the rigorous benchmarks for assessing the true depth of a model's capability, particularly for large language models (LLMs). Unlike tasks driven by pattern recognition, reasoning requires that models infer, generalize, and operate over complex abstract relationships, capabilities that are fundamental to advancing machine intelligence toward human-level performance.

Despite the impressive progress enabled by large-scale pre-training and post-training, current models still fall short of robust reasoning (Guo et al., 2025). Much of the progress to date has come from scaling (Kaplan et al., 2020; Brown et al., 2020; Achiam et al., 2023), such as expanding datasets, model size, and optimization steps. They deliver performance gains but do not explain why reasoning remains intrinsically difficult for machines despite being natural for humans. Another line of

[1]Mohamed bin Zayed University of Artificial Intelligence   [2]Carnegie Mellon University   [3]Stanford University

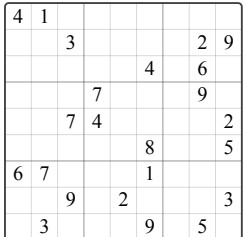
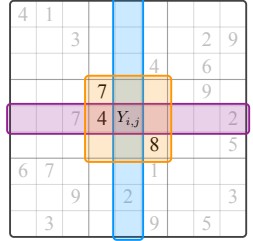
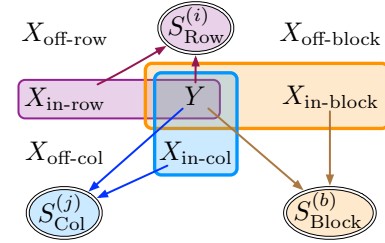

(a) Example Sudoku problem   (b) Single entry in Sudoku   (c) Validity criteria (row, column, block)

Figure 1: **Illustration of reasoning tasks and the selection mechanism, using Sudoku as an example.** (a) A sample $9 \times 9$ Sudoku puzzle with a subset of given clues; the goal is to fill the remaining cells so that each row, column, and $3 \times 3$ subgrid contains the digits 1–9 exactly once. (b) A single unfilled cell $Y_{ij}$ with its row (purple), column (blue), and $3 \times 3$ block (orange) highlighted; the digits within these groups impose constraints that determine the admissible values for $Y_{ij}$. (c) Selection mechanism: $Y$ is valid iff the validity criteria are satisfied $S_{\text{Row}}^{(i)} = S_{\text{Col}}^{(j)} = S_{\text{Block}}^{(b)} = 1$.

work introduces chain-of-thought supervision or human reward to guide models through intermediate reasoning steps (Jaech et al., 2024; Guo et al., 2025; Wang et al., 2025b). However, they often encourage models to mimic the surface form of human explanations, such as natural language, rather than uncover the underlying structure of reasoning in the latent space. As a result, the generated reasoning traces may be verbose, inconsistent, or logically unsound, limiting their value for genuine generalization. This gap highlights the need for a new perspective that goes beyond scaling and imitation, aiming instead to model the structural nature of reasoning itself. The detailed discussion about related work can be found in Appendix A.1.

Specifically, we formulate reasoning tasks as a selection mechanism (Zheng et al., 2024; Elwert & Winship, 2014), a concept from causality that describes how observed variables co-occur under certain selection constraints. In reasoning tasks, high-level logical concepts act as operators that impose constraints on the observed inputs. For example, as illustrated in Figure 1, the numbers in a Sudoku grid must satisfy the validity criteria, where a given entry that needs to be filled should consider the selection constraints such that each row, column, and subgrid contains all digits exactly once. This formulation captures reasoning as the process of narrowing possible latent assignments to those consistent with the selection mechanism.

Building on the selection mechanism formulation, we propose two fundamental hypotheses that further characterize its structure and clarify the sources of difficulty in reasoning. First, the latent space in the selection mechanism is more complex than the observation space, even when the correct answer is uniquely determined by the observed input. For example, when solving a math problem, the observed input may consist of a few numbers and operations with a fixed solution, but the underlying reasoning process involves exploring a vast latent space of intermediate steps, alternative solution paths, and logical transformations before converging to the unique correct answer. Second, the latent variables are densely structured and highly interdependent, making them challenging for current models to disentangle and exploit. This is evident in reasoning processes where a single change in a principle or logical step often necessitates a cascade of adjustments to other steps in order to maintain overall consistency. These properties make it difficult for current models to disentangle and effectively utilize the latent structure, thereby limiting their reasoning capability.

Inspired by this formulation, we propose a new framework $\textbf{SR}^2$ to explicitly model the **S**election, **R**eflection, and **S**elf-**R**efinement, enabling iterative refinement of latent space in the reasoning process. This design encourages the model to explicitly capture the dense dependencies among latent representations, rather than treating them as isolated or independent. This framework consists of three key components. 1) To address the challenge of the intractably large latent space, we propose a reflective representation learning module. This module incorporates the estimated latent variables and maps them back into the input space as feedback injection and iteratively refines the representation. 2) To capture dense dependencies, we propose a dependency self-refinement process that discards observation signals and relies solely on latent variables as inputs to a fixed-point solver, ensuring that the resulting solution satisfies the imposed constraints. 3) To mitigate gradient vanishing in the long iterative refinement process, we introduce periodic intermediate alignment, which injects supervision (or self-supervision) at selected intervals to provide guidance and preserve consistency across reasoning steps.

We validate our approach on canonical reasoning benchmarks, including Sudoku and Maze Navigation. Our method achieves substantial improvements in reasoning accuracy, outperforming the recent hierarchical reasoning model (HRM (Wang et al., 2025a)) by more than 10% while using up to $8\times$ fewer parameters. These results not only highlight the effectiveness of our framework but also demonstrate that reasoning can be advanced through causal modeling of latent structures rather than relying solely on scaling. In addition, we conduct a detailed ablation study to evaluate the contribution of each module, providing insights that may guide future research and development.

In summary, this work contributes:

- A causal perspective that formulates reasoning as selection mechanisms, where concepts act as operators constraining observed inputs;
- The finding of two key properties, including latent space complexity and dense dependencies, that explain the inherent difficulty of reasoning tasks;
- A novel framework with reflective representation learning, dependency self-refinement, and adaptive alignment to model reasoning more effectively;
- Empirical evidence on reasoning benchmarks showing that our approach achieves state-of-the-art performance with far fewer parameters.

We believe this study takes a step toward a deeper understanding of reasoning in machine learning and offers new insights for building models that reason more like humans.

## 2 METHOD

In this section, we present our framework $\mathbf{SR}^2$ (Selection, Reflection, and Self-Refinement), which is designed to address the inherent difficulty of reasoning tasks through a causal formulation. We begin by revisiting reasoning tasks as a selection mechanism, then introduce two hypotheses that explain their complexity, and finally describe our framework components that explicitly model selection, reflection, and self-refinement.

### 2.1 REASONING AS CONSTRAINT-SATISFACTION VIA SELECTION MECHANISMS

In a constraint-satisfaction framing, reasoning is understood as the process of aligning different pieces of information, rules, and procedures so that they cohere with one another. Success, then, is measured less by whether the final outcome is "correct" in some absolute sense, and more by whether the reasoning process maintains internal consistency across the constraints. This makes the characterization of reasoning more about procedural coherence than about reaching a single externally defined answer. Therefore, we revisit reasoning tasks through the lens of causality and formalize them as a selection mechanism. In causal modeling, a selection mechanism describes how observed variables co-occur under certain constraints imposed by latent factors. Analogously, reasoning tasks can be viewed as a process in which high-level logical concepts act as operators that constrain the admissible configurations of the observed inputs.

**Data-generating process as a selection mechanism.** Let $\mathbf{x}$ denote the observed input (e.g., a partially filled Sudoku grid), $\mathbf{y}$ the corresponding solution (e.g., the other entries in Sudoku to be filled), and $\mathbf{z}$ the latent variables representing reasoning rules (e.g., logical principles or arithmetic laws). As shown in Figure 2, we model reasoning as a constrained conditional generation process, where high-level rules $\mathbf{z}$ generate observed pairs $(\mathbf{x}, \mathbf{y})$ only if they satisfy the selection constraints $S(\mathbf{z}) = 1$. Formally, the joint distribution is

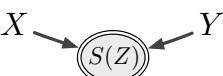

Figure 2: **Causal graph of the selection mechanism where (x,y) are selected by constrains $S(\mathbf{z})$.**

$$p(\mathbf{x}, \mathbf{y}) = \int p(\mathbf{z}) \, p_g(\mathbf{x}, \mathbf{y} \mid \mathbf{z}) \, \mathbb{I}\big(S(\mathbf{z}) = 1\big) \, d\mathbf{z}. \tag{1}$$

where $p(\mathbf{z})$ is the prior over latent rules, $p_g(\mathbf{x}, \mathbf{y} \mid \mathbf{z})$ denotes the base generative distribution parameterized by the function $g$, and $\mathbb{I}(S(\mathbf{z}) = 1)$ enforces that the data satisfies the criteria denoted in the selection variables $\mathbf{z}$. In the deterministic case, $p_g(\mathbf{x}, \mathbf{y} \mid \mathbf{z})$ reduces to a Dirac delta,

$p_g(\mathbf{x}, \mathbf{y} \mid \mathbf{z}) = \delta\big((\mathbf{x}, \mathbf{y}) - g(\mathbf{z})\big)$. Formally specifying the data-generating process provides the theoretical foundation for our subsequent module design; in Section 2.4, we revisit the $\text{SR}^2$ architecture through the lens of data generation.

**Example: Sudoku.** In Sudoku, as illustrated in Figure 1, $\mathbf{x}$ is the initial partially filled grid (1a), $\mathbf{y}$ is other entries to be completed (1b), and $\mathbf{z}$ represents the latent rules which characterizes the relationship between $X$ and $Y$: each row, column, and block (respectively) must contain all digits exactly once (1c). The set of selection mechanisms is given by

$$\mathcal{S} := \big\{ S(\cdot) : \mathcal{Z} \to \{0, 1\} \big\}. \tag{2}$$

where $S$ denotes the selection mechanism and the Boolean value $S(z)$ (or $S_z$) indicates whether the current pair $(X, Y)$ satisfies rule $z$. Thus, solving Sudoku corresponds to selecting $(\mathbf{x}, \mathbf{y})$ pairs (which are derived from mapping $\mathbf{x}$ to the space of numbers) such that the corresponding $\mathbf{z}$ satisfies (a set of) selection criteria $S \in \mathcal{S}$.

Concretely, consider the purple row in Figure 1b. Let $Y_{ij}$ be the number to be filled in the $i$-th row and $j$-th column, and let $X_{\text{in-row}}$ denote all already-filled numbers in the same row. If $Y_{ij}$ is different from every number in $X_{\text{in-row}}$, we set $S^i_{\text{row}} = 1$; otherwise, we set $S^i_{\text{row}} = 0$. Here, $z = \text{Row}$ encodes the rule that each row must contain every digit exactly once; that is, the digits in $X_{\text{in-row}}$ together with $Y_{ij}$ must all be distinct.

**Implication.** This perspective views reasoning as a *selection among latent rules*: the task is solved once $(\mathbf{x}, \mathbf{y})$ is identified under the governing $\mathbf{z}$. Unlike direct input-output mapping, this requires navigating a rule-constrained latent space, highlighting the intrinsic complexity of reasoning tasks.

## 2.2 Two Fundamental Hypotheses

Building on the selection mechanism formulation in Equation 1, we state two hypotheses that formally characterize the sources of difficulty in reasoning tasks.

**Hypothesis 1 (Latent space complexity)** *Let $\mathbf{z}$ denote the latent rules, $(\mathbf{x}, \mathbf{y})$ the observed input–answer pair, and $S(\mathbf{z})$ the selection constraints. Even when the mapping $(\mathbf{x}, \mathbf{y}) = g(\mathbf{z})$ is deterministic for all admissible $\mathbf{z}$ with $S(\mathbf{z}) = 1$, the effective latent space $\mathcal{Z}_S = \{\mathbf{z} : S(\mathbf{z}) = 1\}$ is much larger than the observation space $\mathcal{X} \times \mathcal{Y}$. Thus, a unique observed pair $(\mathbf{x}, \mathbf{y})$ typically corresponds to a large set of possible latent assignments $\mathbf{z} \in \mathcal{Z}_S$ that must be marginalized. For example, solving a math problem with input $\mathbf{x}$ may yield a unique solution $\mathbf{y}$, yet the reasoning process involves exploring a combinatorial number of intermediate derivations in $\mathbf{z}$.*

**Hypothesis 2 (Dense interdependence)** *The admissible latent variables $\mathcal{Z}_S$ exhibit strong structural dependencies. Formally, for $\mathbf{z} = (z_1, \ldots, z_m) \in \mathcal{Z}_S$, the conditional distribution $p(z_i \mid z_{-i}, S(\mathbf{z}) = 1)$ is highly concentrated, indicating that any perturbation in one component $z_i$ typically requires coordinated changes in many other components $z_{-i}$ to preserve validity. For instance, in Sudoku, changing one entry in $\mathbf{z}$ propagates constraints across multiple rows, columns, and subgrids to restore consistency.*

**Implication.** Together, Hypotheses 1 and 2 indicate that reasoning tasks cannot be solved by relying solely on shallow correlations in $(\mathbf{x}, \mathbf{y})$. While the expressive capacity of deep models allows them to fit mappings from $\mathbf{x}$ to $\mathbf{y}$ by learning latent $\mathbf{z}$, such solutions often fail to generalize across tasks. Robust reasoning instead needs the modeling of the complex dependence in the structured latent space $\mathbf{z}$. Hypothesis 1 motivates the need for a reflection mechanism, where models repeatedly explore over-complex latent space and revise latent assignments with feedback. Hypothesis 2 further suggests that even if the mapping from $\mathbf{x}$ to $\mathbf{z}$ is learned, the dense interdependencies within $\mathbf{z}$ require iterative refinement to reach a consistent solution.

## 2.3 The $\text{SR}^2$ Framework

Based on the implications of our formulation, we propose the $\text{SR}^2$ framework, which implements the **Selection** Mechanism, **Reflection**, and **Self-Refinement** modules. Our approach encourages

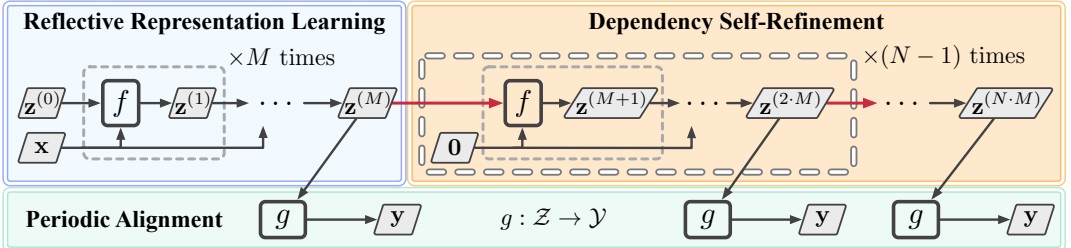

Figure 3: **Overall framework of the SR$^2$ method.** The framework consists of three main modules: Reflective Representation Learning (in blue), Dependency Self-Refinement (in orange), and Periodic Alignment (in green). $f$ denotes the weight-shared atomic block that updates the latent state, and $g$ projects the latent space to the final answers. In the representation learning stage, $f$ recurrently updates the latent state with the observation as injection, $\mathbf{z}^{(t+1)} = f(\mathbf{z}^{(t)}, \mathbf{x})$, for $M$ steps to obtain a refined initialization. Next, in the self-refinement stage, the model drops the observation signal, $\mathbf{z}^{(t+1)} = f(\mathbf{z}^{(t)}, \mathbf{0})$, and updates for a long $M \times (N-1)$ steps to resolve dense dependencies and approach a fixed point. Throughout training, the supervision is conducted periodically (e.g., every $M$ steps) to stabilize long recurrences and mitigate gradient vanishing. When adding supervision, the gradients from future states are blocked (as the red arrows).

the model to iteratively refine latent variables, capture dense dependencies, and enforce consistency across reasoning steps. The pipeline can be found in Figure 3, which is composed of three modules: Reflective Representation Learning, Dependency Self-Refinement, and Periodic Alignment.

**Reflective Representation Learning.** A key challenge posed by Hypothesis 1 is the intractability of the large latent space $\mathbf{z}$, which makes direct inference difficult. To address this issue, we propose a reflective representation learning module, which explicitly models reasoning as a reflection process.

We define reflection as a fixed-point equation of the form

$$\mathbf{z} = f(\mathbf{z}, \mathbf{x}), \tag{3}$$

where $f(\cdot)$ is a parametric function (e.g., Transformer model) that updates the latent representation $\mathbf{z}$ conditioned on the observed input $\mathbf{x}$. This formulation captures the intuition that reasoning is not a one-shot mapping from $\mathbf{x}$ to $\mathbf{z}$, but rather an iterative process in which $\mathbf{z}$ is repeatedly revised until it becomes consistent with both the observed data and the latent constraints. Concretely, equation 3 formalizes reflection by refining the latent solution $\mathbf{z}$ through fixed-point updates rather than producing it in one step. This directly addresses Hypothesis 1: when the latent space far exceeds the observation space, iterative fixed-point updates provide a practical route to contract toward a constraint-satisfying $\mathbf{z}$.

Equation equation 3 can be solved by iterative refinement:

$$\mathbf{z}^{(n+1)} = f(\mathbf{z}^{(n)}, \mathbf{x}), \quad n = 0, 1, \ldots, N, \tag{4}$$

starting from an initial estimate $\mathbf{z}^{(0)}$ initialized as zeros. After $T$ iterations, the fixed-point solution $\mathbf{z}^*$ is expected to approximate the reflective latent representation satisfying equation 3. This iterative mechanism enables the model to explore the latent space incrementally, progressively eliminating invalid assignments and converging towards rule-consistent solutions.

In large latent spaces, explicit conditioning on the previous state is often essential for guiding inference. Although direct mapping from $\mathbf{x}$ to $\mathbf{z}$ is mathematically equivalent to reflection, it fails to sufficiently constrain the solution space. Appendix B provides a motivating example showing that iterative conditioning yields strictly better solutions by progressively reducing the solution space.

**Flattened recurrent Transformer.** While the reflective update function $f(\mathbf{z}, \mathbf{x})$ in equation 3 can in principle be instantiated by any deep neural network, such as a standard Transformer, we take particular inspiration from recent advances in Deep Equilibrium Models (DEQ) (Bai et al., 2019), which establish a duality between model depth and iterative refinement.

Consider a standard Transformer model with $L$ stacked layers, where each layer applies the same structural transformation to its input representation. Formally, let $h^{(l)}$ denote the hidden state at layer $l$. The forward propagation can be expressed as

$$h^{(l+1)} = \mathcal{T}^l(h^{(l)}, \mathbf{x}), \quad l = 0, 1, \ldots, L-1, \tag{5}$$

where $\mathcal{T}(\cdot)$ is a Transformer block. Instead of explicitly stacking $L$ distinct layers, we propose to *flatten* the architecture by reusing a single Transformer block $\mathcal{T}$ recurrently across iterations:

$$h^{(m+1)} = \mathcal{T}(h^{(m)}, \mathbf{x}), \quad m = 0, 1, \ldots, M, \tag{6}$$

This unrolled architecture can be viewed as solving a fixed-point problem, where repeated applications of $\mathcal{T}$ iteratively refine the latent state. Equivalently, a depth-$L$ Transformer can be reinterpreted as a depth-one recurrent Transformer unrolled for $M = L$ steps. By replacing $M$ distinct layers with a single shared module, the model reduces parameters while preserving expressiveness through iteration. Such aligned iterative refinement allows the model to achieve effective depths much greater than its deep explicit architecture.

Unlike DEQ, which seeks an equilibrium state and computes gradients via the implicit function theorem without storing intermediate iterates, we explicitly unroll a fixed number of refinement steps. In this sense, our approach is a truncated, explicitly unrolled variant that trains the refinement trajectory itself rather than directly optimizing an implicit fixed point.

**Dependency Self-Refinement.** Hypothesis 2 highlights that the latent variables $\mathbf{z}$ are densely structured and strongly interdependent. A change in one latent component typically requires a cascade of adjustments to other components in order to maintain global consistency. Therefore, a reasoning model needs to capture and refine such interdependencies, rather than relying solely on direct mappings from the observed input $\mathbf{x}$.

To model these dependencies, we propose a self-refinement process in which the input signal $\mathbf{x}$ is removed, and the latent dynamics evolve autonomously:

$$\mathbf{z}^{(t+1)} = f_s(\mathbf{z}^{(t)}, \mathbf{0}), \quad t = 0, 1, \ldots, T, \tag{7}$$

where $\mathbf{z}^{(0)}$ is initialized from the reflective representation learning stage, and $f_s(\cdot)$ is an iterative solution of the fixed-point operator defined on the latent space. This formulation forces the model to refine $\mathbf{z}$ using only its internal structure, ensuring that the converged solution satisfies the logical constraints implied by the selection mechanism. By eliminating direct dependence on $\mathbf{x}$, the model is discouraged from learning spurious shortcuts and instead is compelled to resolve latent dependencies through iterative updates.

Experimentally, we found that both reflective representation learning and dependency self-refinement can be realized by a single atomic module, such as a Transformer block $f_s() = f()$, reused across iterations. This design substantially reduces the parameter footprint while maintaining competitive performance. Finally, we repeat a Transformer block with $T = M \times N$ times and re-order them, where the first $M$ iterations inject $\mathbf{x}$ for representation learning, and the last $(M - 1) \times N$ iterations are used for dependency learning.

**Periodic Alignment.** With the flattened recurrent model, we have $T = M \times N$ iterations. Such long iterative refinement processes often suffer from optimization challenges. In particular, gradients propagated through many iterations tend to vanish, leading to unstable training and poor convergence. To address this challenge, we propose a periodic alignment mechanism. Instead of applying supervision only at the final step $T$, we inject additional supervision (or self-supervision) signals at regular intervals throughout the iterative refinement process. Formally, for refinement states $\{\mathbf{z}^{(t)}\}_{t=1}^{T}$, the training objective is augmented as

$$\mathcal{L} = \sum_{t \in \mathcal{A}} \ell(g(\mathbf{z}^{(t)}), \mathbf{y}), \tag{8}$$

where $g(\cdot)$ is a prediction head mapping latent variables to task outputs, $\ell(\cdot, \cdot)$ is a task-specific loss function, and $\mathcal{A} \subseteq \{1, \ldots, T\}$ specifies the alignment intervals. Typical choices of $\mathcal{A}$ include uniform spacing (e.g., every $M$ steps) or adaptive spacing based on convergence criteria. To ensure stability, we detach the computational graph at each alignment step, so that each iteration only receives near gradients from its own supervision signal rather than from far future steps. This periodic alignment stabilizes optimization by providing intermediate guidance that mitigates gradient vanishing and enforces consistency, ensuring that latent states $\mathbf{z}^{(t)}$ remain progressively aligned with the target throughout the refinement process.

## 2.4 CONNECTING SR2 TO THE DATA GENERATING PROCESS

**From Eq. equation 1 to the target.** Equation equation 1 describes the data generating process for our task rather than the model design. In implementation, SR2 learns the conditional distribution $p(\mathbf{y} \mid \mathbf{x})$ instead of the joint $p(\mathbf{x}, \mathbf{y})$, and this conditional is derived from the joint as

$$p(\mathbf{y} \mid \mathbf{x}) = \int p(\mathbf{z} \mid \mathbf{x}) \, p_g(\mathbf{y} \mid \mathbf{x}, \mathbf{z}) \, \mathbb{I}\big(S(\mathbf{z}) = 1\big) \, d\mathbf{z}, \tag{9}$$

**Implementation map in SR2.** **Latent inference** $p(\mathbf{z} \mid \mathbf{x})$. A single Transformer layer produces a latent state $\mathbf{z}$. SR2 refines it through reflection updates $\mathbf{z}^{t+1} = f(\mathbf{x}, \mathbf{z}^t)$, yielding the final state used for prediction. **Prediction head** $p_g(\mathbf{y} \mid \mathbf{x}, \mathbf{z})$. Since $\mathbf{z}$ summarizes $\mathbf{x}$, a `lm_head` is trained to model $p(\mathbf{y} \mid \mathbf{z})$ as a surrogate. **Selection** $\mathbb{I}\big(S(\mathbf{z}) = 1\big)$. Training pairs $(\mathbf{x}, \mathbf{y})$ satisfy the task rules, so SR2 concentrates probability on rule consistent states without an explicit selector.

## 3 EXPERIMENTS

### 3.1 BENCHMARKS

**Sudoku-Extreme** The Sudoku task requires filling a 9×9 grid with the digits 1–9 such that each row, each column, and each 3×3 subgrid contains all digits without repetition. Due to the logical complexity of its constraints, Sudoku is a widely used benchmark for evaluating the logical reasoning ability of machine learning models (Palm et al., 2017; Long, 2023; Du et al., 2024). For a fair comparison, we use the same dataset, Sudoku-Extreme, and follow the setup strictly with the recent SOTA method, HRM (Wang et al., 2025a). Specifically, the benchmark comprises limited 1,000 training puzzles and 422,786 test puzzles, with an average difficulty of 22 backtracking steps.

**Maze-Hard** The maze navigation task asks a model to find the optimal path between a designated start and goal within a 30×30 maze. The requirement to plan long, intricate paths makes it an effective probe of a model's reasoning ability (Darlow et al., 2025; Su et al., 2025; Lehnert et al., 2024b). We use the Maze-Hard dataset released by (Wang et al., 2025a), which follows the instance-generation procedure of (Lehnert et al., 2024a). The generated mazes are filtered to retain only the hardest cases, i.e., those whose shortest paths are the longest(Kapadia et al., 2013). The benchmark comprises 1,000 training instances and 1,000 test instances.

**ARC AGI** The Abstraction and Reasoning Corpus for Artificial General Intelligence (ARC AGI) evaluates abstract pattern induction and compositional generalization via small grid–transformation tasks that require minimal prior knowledge and probe fluid intelligence (Chollet, 2019). Each task provides a few input–output demonstrations; the solver must produce exactly matching outputs for novel inputs. Following the official protocol, we use ARC AGI 1 (Chollet, 2024) and ARC AGI 2 (ARC Prize Foundation, 2025): ARC AGI 1 has 400 training and 400 evaluation tasks; ARC AGI 2 has 1,000 public training and 120 public evaluation tasks. We adopt a two-trial success criterion and report pass@2 accuracy.

### 3.2 BASELINE AND IMPLEMENTATION

We compare our $\mathbf{SR}^2$ framework with several canonical network architectures used for reasoning, as well as the recent HRM (Wang et al., 2025a) model, which are summarized below:

- **Standard Transformer** (Vaswani et al., 2017). Implements a standard multi-layer Transformer architecture.
- **Block Universal Transformer** (Dehghani et al., 2018). Replaces a deep stack of Transformer blocks with a flat recurrent architecture that repeatedly applies a Transformer block (a single Transformer layer in the experiments)
- **Recurrent Depth** (Geiping et al., 2025b). Applies the same Transformer layer iteratively, while additionally mapping each layer's latent representation back to the input space. In other words, each unit is injected by both inputs and latent states.

Table 1: **Main results comparing SR$^2$ with baselines on Sudoku-Extreme, Maze-Hard and ARC-AGI benchmarks.** We report pass@1 accuracy (%) and the number of learnable parameters ("Params"). With only 3.4M parameters ( 1/8 of the 27.3M Transformer), SR$^2$ achieves the best accuracy on both tasks (best in bold).

| Method | Params | Sudoku-Extreme | Maze-Hard | ARC-1 | ARC-2 |
|---|---|---|---|---|---|
| Transformer | 27.3M | 1.17 | 0 | 21.0 | 0 |
| Block Universal Transformer | 3.4M | 0 | 30.4 | - | - |
| Recurrent Depth | 3.4M | 42.52 | 48.4 | - | - |
| HRM | 27.3M | 55.0 | 74.5 | 40.3 | 5.0 |
| Reflective Model | 27.3M | 53.12 | 70.8 | - | - |
| SR$^2$ | **3.4M** | **66.63** | **93.7** | **44.3** | **6.7** |

- **HRM** (Wang et al., 2025a). HRM employs an interactive recurrent structure, with a high-level module responsible for abstract planning and a low-level module dedicated to detailed computation. For implementation, we follow the officially released HRM configurations.

- **Reflective Model** (Our baseline model). In this baseline, we consider only the reflective representation learning stage, excluding the subsequent self-refinement process. Each unit is implemented as an 8-layer Transformer without flattening, equipped with both input injection and supervision.

For fairness, all methods are trained with the same optimizer, learning rate, and batch size. Parameter counts are matched across similar designs: the Transformer, Reflective Model, and HRM uses 8 Transformer layers (4 high-level layers, 4 low-level layers in HRM), while the Block Universal Transformer, Recurrent Depth, and our **SR$^2$** model adopt a single Transformer layer as the backbone. More discussions with other recurrent models can be found in Appendix A.1.

## 3.3 EXPERIMENTAL RESULTS

We report the performance and parameter counts of **SR$^2$** and competing methods on the *Sudoku-Extreme* and *Maze-Hard* benchmarks, as summarized in Table 1. For ARC-1 and ARC-2, due to computational resource limitations, we did not evaluate all baseline models. Several key observations emerge:

- **Standard Transformer.** Standard Transformer architectures fail to solve these demanding reasoning tasks. Even with extended training, the model does not acquire the necessary reasoning skills, underscoring the importance of reflective mechanisms.

- **Block Universal Transformer.** Flattening the architecture by reusing a single Transformer block yields modest gains over the standard Transformer, but overall performance remains unsatisfactory.

- **Recurrent Depth.** Adding input injections to the recurrent architecture leads to clear improvements, demonstrating the benefit of feeding intermediate states back into the model.

- **Reflective Model.** Further improvements are observed when both input injections and supervisory signals are balanced. The Reflective Model outperforms Recurrent Depth, highlighting the importance of combining multiple feedback pathways.

- **HRM vs. Reflective Model.** HRM achieves results comparable to the Reflective Model, suggesting that hierarchical recurrent structures are not strictly necessary for reasoning tasks of this type.

- **SR$^2$.** Compared with Recurrent Depth, **SR$^2$** achieves substantial improvements such as over 20% on *Sudoku-Extreme* and nearly 2× on *Maze-Hard*. This demonstrates the crucial role of self-refinement. Moreover, **SR$^2$** surpasses HRM by 11.6% on *Sudoku-Extreme* and 19.2% on *Maze-Hard*, while using only one eighth of HRM's parameters.

## 3.4 TRAINING & INFERENCE COST ANALYSIS

We evaluate the computational efficiency of **SR$^2$** against Transformer and HRM baseline on Sudoku Extreme under same settings, reporting training speed in batches per second, training memory as the

average per GPU across devices, and inference speed as samples processed per second computed from the wall clock runtime on the test set. The Periodic Alignment module makes $\mathbf{SR}^2$ slower than Transformer baseline during both training and inference, yet $\mathbf{SR}^2$ remains faster than HRM because it uses a single layer without a hierarchical structure. $\mathbf{SR}^2$ also consumes somewhat more training memory than the other baselines.

## 3.5 ABLATION STUDIES

We perform a comprehensive set of ablation studies to systematically evaluate the contribution of each component. **No Self-Refinement** removes Dependency Self-Refinement; all $T = M \times N$ iterations inject input $x$ and do not explicitly model dependencies among latent $z$. **No Reflection** removes Reflective Representation Learning process; $x$ is injected only at the first iteration (no repeated injections across the first $M$). **Mixture (2/4 Reflection)** performs

Table 2: **Training and inference efficiency comparison**.

| Method | Training Speed (Batch/s) | Training Mem (GB) | Inference Speed (Sample/s) |
|---|---|---|---|
| Direct Pred | 21.39 | 3.024 | 7489.6 |
| HRM | 10.57 | 3.231 | 1487.7 |
| $SR^2$ | 14.73 | 3.950 | 2073.6 |

multiple self-refinement phases by re-injecting $x$ at evenly spaced blocks rather than only at the first block. **Separate Function** uses two distinct parameterized functions for the reflective and self-refinement stages instead of a shared one. **Flattened Reflective Model** replaces the multi-layer stack of the Reflective Model with a single recurrent Transformer layer, unrolled across iterations.

**Is every component of $\mathbf{SR}^2$ necessary? Yes.** To assess the roles of the Reflective Representation Learning and Dependency Self-Refinement modules, we conduct ablations that remove each module in turn. In *No Self-Refinement*, the entire self-refinement loop is replaced by a single feedback injection. In *No Reflection*, we remove the feedback step from each alignment, i.e., no input injection is performed. As reported in Table 3, eliminating the self-refinement loop prevents the model from capturing dependencies and leads to a clear drop in accuracy, while removing feedback al-

Table 3: **Ablation of the $SR^2$ framework**.

| Ablations | Accuracy (%) |
|---|---|
| No Self-Refinement | 53.11 |
| No Reflection | 0 |
| Mixture (2 Reflections) | 63.32 |
| Mixture (4 Reflections) | 55.25 |
| Separate Function | 59.76 |
| Reflective Model | 53.12 |
| Flattened Reflective Model | 53.75 |
| $SR^2$ | **66.63** |

together leaves the model unable to retain input features, causing performance to collapse to zero.

**Does more reflections feedback always help? No.** We modify the default $\mathbf{SR}^2$ design (only one reflective representation learning block in M $\times$ N iterations) to test using two and four reflections. As reported in Table 3, Mixture (2 Reflections) produces a slight decrease in accuracy relative to the default, while Mixture (4 Reflections) suffers a large degradation. These results indicate that injecting input too frequently biases the model toward fitting superficial input patterns and distracts it from learning deeper dependencies among latent states. In practice, a single reflection per alignment offers the best trade-off for $\mathbf{SR}^2$.

**How should m and n be chosen under a fixed budget?** As Figure 4 shows, accuracy peaks when $M \approx N$ and declines as their ratio departs from 1. The decline is asymmetric: settings with $M > N$ degrade more gently than their mirrored $N > M$ counterparts, so if imbalance is unavoidable, favor a larger $M$. The results indicate that balanced configurations are preferable.

**Does the Reflection and Self-Refinement modules have to learn two distinct functions? No.** We replace the single shared function in $\mathbf{SR}^2$ with two Transformer layers that do not share parameters, assigning one to the Reflection module and the other to the Self-Refinement module. As shown in Table 3, the decoupled design Separate

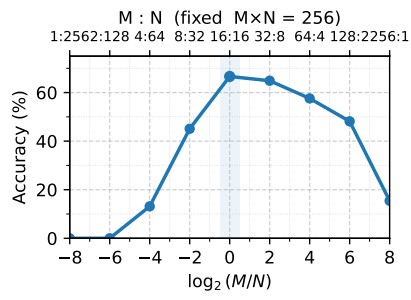

Figure 4: **Choosing $N$ and $M$ under a fixed compute budget.**

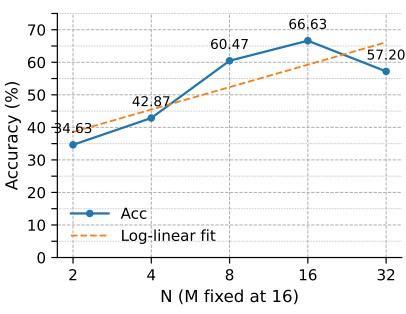 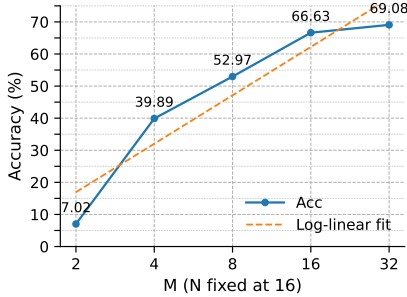

(a) Accuracy vs. $N$ with $M=16$ fixed.     (b) Accuracy vs. $M$ with $N=16$ fixed.

Figure 5: **Accuracy scaling with the hyperparameter $N$ and $M$**: the left panel varies $N$ at $M=16$, and the right varies $M$ at $N=16$.

Function even reduces accuracy. These results suggest that a single unified function is sufficient to support both representation learning and refinement.

**Is the flat architecture effective? Yes.** To test whether a recurrent single-layer Transformer can replace a stack of distinct layers in our framework, we compare *Reflective Model* with *Flattened Reflective Model*, where the former uses eight distinct Transformer layers, whereas the latter applies a single Transformer layer recurrently with shared weights. The results show a comparable accuracy between the two, indicating that parameter sharing in depth can serve as a drop-in alternative to a deeper stack without degrading reasoning performance.

**How does performance scale with the hyperparameter $M$ and $N$?** Using the results in Figure 5, we observe a near-monotonic improvement as $M$ increases with $N$ fixed at 16, well captured by a log-linear trend. Holding $M=16$ and sweeping $N$, the precision increases to a rapid peak before dropping, indicating early saturation and eventual regression when $N$ becomes too large. Although larger $M$ and moderately large $N$ yield stronger results, they also inflate training time and slow inference. To balance accuracy with computational cost, we adopt $M=N=16$ as the default setting for our main experiments.

**How many alignment steps should we use at test time?** As shown in Figure 6, reducing the test-time alignment horizon steps monotonically lowers accuracy but increases throughput (samples / s), trace a clear accuracy-speed Pareto frontier over $steps \in \{32, 16, 8, 4, 2\}$. Notably, for a model trained with $M=16$, evaluating with a larger steps (e.g., 32) does not yield additional accuracy gains, indicating that the training-time horizon effectively caps test-time benefits. In practice, steps can be tuned at deployment to satisfy latency budgets with a predictable accuracy cost, whereas pushing steps beyond the training setting offers little return.

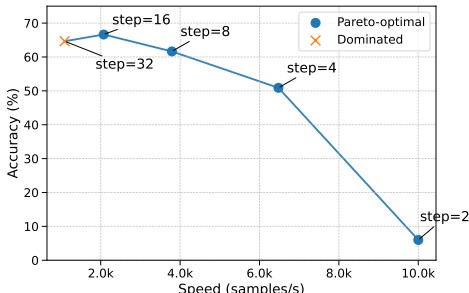

Figure 6: **Accuracy–throughput trade-off vs. test-time alignment steps.**

## 4 CONCLUSION

We revisited reasoning tasks through the lens of causality and formulated them as a selection mechanism, identifying latent space complexity and dense interdependence as key sources of difficulty. Building on these insights, we introduced **SR**$^2$, a framework that models reasoning as iterative refinement with three components: reflective representation learning, dependency self-refinement, and periodic intermediate alignment, instantiated by a flat recurrent Transformer. Experiments on Sudoku and Maze Navigation benchmarks demonstrate that SR$^2$ achieves significant gains in reasoning accuracy while using up to $8\times$ fewer parameters than recent baselines. **Limitations:** Despite its insight, the assumption of reasoning as a selection mechanism may oversimplify the richness of human reasoning. Besides, our evaluation is limited to benchmarks with clear and simple logical constraints, and we leave more comprehensive evaluations on complex or open-domain tasks, as well as large-scale implementations, to future work.

ACKNOWLEDGMENT

We would also like to acknowledge the support from NSF Award No. 2229881, AI Institute for Societal Decision Making (AI-SDM), the National Institutes of Health (NIH) under Contract R01HL159805, and grants from Quris AI, Florin Court Capital, MBZUAI-WIS Joint Program, and the Al Deira Causal Education project.

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

# Appendix for "Selection, Reflection and Self-Refinement: Revisit Reasoning Tasks via a Causal Lens"

## A  MORE DETAILS OF PAPER

### A.1  RELATED WORK

**Reasoning in Machine Learning**   Reasoning, the process of deriving conclusions from information through logical inference, has long been viewed as a central challenge for artificial intelligence. Classical approaches in symbolic AI treated reasoning as explicit rule-based deduction, but these systems struggled with scalability and generalization (McCarthy, 1988; Russell & Norvig, 2010). In contrast, recent advances in machine learning have shifted toward data-driven reasoning, where models learn to approximate reasoning behaviors from large-scale corpora. Large language models (LLMs) such as GPT-o1 (Jaech et al., 2024), and Gemini (Team et al., 2023) have demonstrated impressive progress on reasoning tasks, benefiting from advances in scaling post-training. Methods such as chain-of-thought supervised finetuning and reinforcement learning optimization provide improvements (Guo et al., 2025; Wang et al., 2025b; Yue et al., 2025; Xie et al., 2025).

**Recurrent Latent Reasoning**   Another line of research relevant to our work is recurrent latent reasoning, where reasoning is modeled as an iterative process over hidden states or latent representations. Unlike single-pass architectures, these methods refine intermediate representations through recurrence or iterative updates, aiming to capture long-range dependencies and structured constraints. Classical instances include recurrent relational networks (Palm et al., 2018), which iteratively propagate messages between entities to perform relational reasoning. More recent work, such as Universal transformers (Dehghani et al., 2018), Algoformer (Gao et al., 2024), and Looped Transformers (Giannou et al., 2023) Furthermore, CoTFormer (Mohtashami et al., 2023) and Recurrent Depth (Geiping et al., 2025a) enhance dependency modeling by injecting intermediate representations and original representations into each recurrent unit, respectively. The recent HRM model (Wang et al., 2025a) introduces a hierarchical recurrent structure, where a high-level module is responsible for slow, abstract planning, while a low-level module manages fast, fine-grained computations. Our work shares this motivation of recurrent refinement but differs in two key aspects. First, we formulate reasoning as a selection mechanism grounded in causality, which provides a principled explanation for the difficulty of reasoning tasks. Second, we propose a new recurrent refinement framework that incorporates an early reflection module with dense input injection, a long dependency self-refinement procedure, and periodic intermediate alignment.

**Selection Mechanisms in Causality**   Within causal inference, selection mechanisms describe how observed variables co-occur under specific constraints (Pearl, 2009). This perspective has proven valuable in understanding selection bias and identifiability. Recent work (Deng et al., 2025; Zheng et al., 2024) extends this concept beyond selection bias and shows that sequential data, such as natural language, music, and poems, inherently follow this mechanism. In this paper, we show that the reasoning task also fits this mechanism, where high-level logical concepts act as operators imposing constraints on observed inputs.

**Fixed-Point Solutions for Structured Prediction**   Fixed-point methods have been widely applied in optimization and equilibrium modeling, where the solution is defined as a stable point under repeated updates (Bai et al., 2019; 2020; El Ghaoui et al., 2021; Li et al., 2022; Kawaguchi, 2021). This paradigm departs from explicit layer stacking in conventional deep networks (He et al., 2016) or Transformers (Vaswani et al., 2017) and instead characterizes representations as equilibria of nonlinear transformations. Deep Equilibrium Model (DEQ)(Bai et al., 2019; 2020) directly solves for the equilibrium point of a single-layer transformation using root-finding methods, thereby representing infinitely deep networks with constant memory cost. In parallel, the broader field of implicit deep learning (El Ghaoui et al., 2021; Kawaguchi, 2021) has established theoretical guarantees for convergence and optimization, highlighting the potential of implicit formulations to provide compact yet powerful representations. Besides, iterative refinement methods (Geiping et al., 2025a) have also demonstrated effectiveness in reasoning and structured prediction tasks.

**Verbal self-correction in LLMs.**  A growing body of work investigates explicit self-correction for LLMs via verbal feedback, critique, and planning. Reflexion Shinn et al. (2023) introduces verbal self-reflection and episodic memory to iteratively improve task success; Self-Refine Madaan et al. (2023) performs critique-and-edit loops using model-generated feedback; Self-Consistency Wang et al. (2022) reduces reasoning errors by sampling and aggregating multiple chains of thought; STaR Zelikman et al. (2022) bootstraps rationales through iterative data augmentation and fine-tuning; and Constitutional AI Bai et al. (2022) uses a set of principles to guide model-written critiques and refine outputs. While related in spirit, our approach differs in that $SR^2$ does not explicitly generate token sequences to steer self-correction. Instead, the correction process is implemented implicitly through iterative updates of the latent variables $(z)$, integrating self-correction into the latent reasoning dynamics without additional verbal overhead.

## A.2  COUNTING THE LATENT AND OBSERVATION SPACES IN SUDOKU

**Setup and counting argument.**  Consider a standard $9 \times 9$ Sudoku. Let $C = \{1, \ldots, 81\}$ be the index set of cells and let $K \subset C$ be the indices of revealed entries in the initial grid. Define $M = C \setminus K$ as the masked cells with $|M| = n$. A full solution is an assignment $z \in [9]^C$ that satisfies all Sudoku constraints on rows, columns, and $3 \times 3$ blocks and that agrees with the revealed entries on $K$. Let $Z$ be the set of full solutions consistent with the given clues and write $S := |Z|$.

The observation for a fixed instance is the single partially filled grid $x^*$, hence the observation space is a singleton.

$$|\mathcal{O}| = 1.$$

We count the latent objects that we call valid solution trajectories. Starting from the initial grid and for each step $t = 1, \ldots, n$ we choose one still empty cell $c_t \in M \setminus \{c_1, \ldots, c_{t-1}\}$ and fill it with the digit $z(c_t)$ taken from some full solution $z \in Z$. The partial assignment after every step must satisfy all Sudoku constraints.

*Prefix feasibility.* Fix any $z \in Z$. For any subset $T \subseteq M$, the partial assignment $\{(c, z(c)) : c \in T\}$ satisfies all Sudoku constraints, since $z$ already assigns pairwise consistent digits to every row, column, and block, hence any restriction of $z$ remains feasible.

By prefix feasibility, if we commit to one fixed $z \in Z$ then every ordering of the $n$ masked positions yields a valid trajectory. Therefore for a fixed full solution $z$ the number of valid trajectories equals the number of permutations of $M$.

$$|\mathcal{L}(z)| = n!.$$

Trajectories coming from different full solutions are disjoint when viewed as ordered sequences of cell and digit pairs. Indeed, if two ordered sequences coincide then the underlying assignments agree on all masked and revealed cells, hence the two full solutions are the same. Summing over all full solutions gives the size of the latent space.

$$|\mathcal{L}| = S\, n!.$$

**Asymptotics and comparison.**  Using Stirling approximation we obtain

$$n! \ = \ \sqrt{2\pi n}\left(\tfrac{n}{e}\right)^n(1 + o(1)) \ = \ \exp\!\big(\Theta(n \log n)\big).$$

Thus for fixed board size and any fixed $S$ in particular $S = 1$ for uniquely solvable puzzles the latent space grows as

$$|\mathcal{L}| \ = \ \Theta\!\Big(S\,\sqrt{n}\left(\tfrac{n}{e}\right)^n\Big), \qquad \frac{|\mathcal{L}|}{|\mathcal{O}|} \ = \ \Theta(S\,n!).$$

This formalizes that the latent space is much larger than the observation space for fixed instances.

**Remark on stricter move policies.**  If trajectories are restricted to fill only logically forced cells at each step then the admissible orders are the linear extensions of a problem dependent partial order on $M$. Exact counting is difficult, but the bounds below always hold, which preserves the factorial upper bound.

$$1 \ \le \ |\mathcal{L}_{\text{forced}}| \ \le \ n!.$$

## A.3 THEOREM (EXISTENCE OF CONVERGENCE IN $\text{SR}^2$).

Let $f : \mathcal{Z} \times \mathcal{X} \to \mathcal{Z}$ be the shared update function used in both the **Reflection** and **Self-Refinement** stages of $\text{SR}^2$. Define the update rules:

$$\textbf{Reflection:} \quad z^{t+1} = f(z^t, x), \quad t = 0, \dots, M-1,$$
$$\textbf{Self-Refinement:} \quad z^{t+1} = f(z^t, 0), \quad t \geq M.$$

**(A1) Completeness** The latent space $(\mathcal{Z}, \|\cdot\|)$ is a complete metric space, and for any fixed input $x \in \mathcal{X}$, $f(\cdot, x)$ maps $\mathcal{Z}$ to itself.

**(A2) Contraction** There exists a constant $L \in (0,1)$ such that

$$\|f(z_1, x) - f(z_2, x)\| \leq L \|z_1 - z_2\|. \tag{10}$$

for all $z_1, z_2 \in \mathcal{Z}$ and $x \in \{x, 0\}$.

Then:

(i) Each update function $f(\cdot, x)$ and $f(\cdot, 0)$ admits a unique fixed point $z_x^\star$ and $z_0^\star$, respectively.

(ii) The iterative sequences defined by Reflection and Self-Refinement converge to their corresponding fixed points:

$$z^t \to z_x^\star \quad \text{for } t < M, \quad \text{and} \quad z^t \to z_0^\star \quad \text{for } t \geq M.$$

**Proof.** **Step 1. Define the contraction operators.** Define two operators on the latent space: $T_x(z) := f(z, x), \quad T_0(z) := f(z, 0)$. By assumption (A1), both $T_x$ and $T_0$ map $\mathcal{Z}$ into itself.

**Step 2. Show that $T_x$ and $T_0$ are contractions.** By (A2),

$$\|T_x(z_1) - T_x(z_2)\| = \|f(z_1, x) - f(z_2, x)\| \leq L \|z_1 - z_2\|,$$

and similarly for $T_0$. Hence, both $T_x$ and $T_0$ are contraction mappings on the complete metric space $(\mathcal{Z}, \|\cdot\|)$.

**Step 3. Apply the Banach fixed-point theorem Kreyszig (1978).** By Banach's theorem, every contraction on a complete metric space has a unique fixed point, and successive applications of the map converge to it. Thus, there exist unique points $z_x^\star, z_0^\star \in \mathcal{Z}$ such that

$$T_x(z_x^\star) = z_x^\star, \quad T_0(z_0^\star) = z_0^\star.$$

Moreover, for any initialization $z^0 \in \mathcal{Z}$,

$$z^{t+1} = T_x(z^t) \implies z^t \to z_x^\star, \quad z^{t+1} = T_0(z^t) \implies z^t \to z_0^\star.$$

**Step 4. Apply the result to $\text{SR}^2$ dynamics.** In $\text{SR}^2$, the reflection phase iterates $T_x$ for a finite number of steps ($M$). Therefore, after sufficiently many steps, the latent variable $z^M$ approaches $z_x^\star$. The self-refinement phase then iterates $T_0$ starting from $z^M$, so by Banach's theorem, $z^t \to z_0^\star$ as $t \to \infty$.

**Conclusion.** Under mild contraction and completeness assumptions (standard in implicit and equilibrium formulations) the $\text{SR}^2$ updates possess unique equilibria for both reflection and self-refinement, and their iterative applications converge to these equilibria. The theorem guarantees existence of convergence for the latent representations and therefore for the predictions of $\text{SR}^2$. $\square$

## B  A MOTIVATING EXAMPLE

We have thoroughly discussed in the main text the necessity of incorporating latent variable recurrence into the update process. Whether through a selection mechanism that guides the representation, or via a fixed-point iterative formulation, the current arguments focus on the existence of a

latent optimal state $z^*$ such that the iterative update $z^{(t)} = f(x, z^{(t-1)})$ converges, in which case the introduction of last latent state $z^{(t-1)}$ into the estimation of $z^{(t)}$ matters.

However, one critical question remains unresolved: why is it insufficient—or fundamentally difficult—to learn a high-quality latent representation $z$ using only the observation $x$, i.e., through a direct mapping $z = g(x)$? Towards this question, we consider a simple yet illustrative example using a linear state-space model:

$$
\begin{aligned}
z^{(1)} &\in N(0, P_0), \\
z^{(t)} &= Az^{(t)} + w_t, \quad A \in \mathbb{R}^{d \times d}, \quad w_t \sim N(0, Q), \\
x^{(t)} &= Cz^{(t)} + v_t, \quad C \in \mathbb{R}^{m \times d}, \quad v_t \sim N(0, R),
\end{aligned}
\tag{11}
$$

with $Q \succ 0, R \succ 0$, and the observation is under-determined $m < d$.

Our objective is to estimate the hidden state $z^{(1:T)}$ from the observation $x^{(1:T)}$ only. We compare two formulations: (1) estimating $z^{(t)}$ using only $x^{(t)}$, and (2) estimating $z^{(t)}$ using both $x^{(t)}$ and $z^{(t-1)}$. We demonstrate that modeling the explicit dependence on $z^{(t-1)}$ constrains the solution space and improves the conditioning of the estimation problem. In contrast, estimating $z^{(t)}$ directly from $x^{(t)}$ alone is generally ill-posed.

Specifically, the second formulation—by incorporating $z^{(t-1)}$—forms a recursive structure that leads to a unique maximum a posterior (MAP) estimate under mild assumptions. It leverages both the observation likelihood and the latent dynamics prior, thereby regularizing the solution and resolving ambiguity. In contrast, the first formulation estimates $z^{(t)}$ solely from $x^{(t)}$, and thus its solution lies on an affine subspace determined by the null space of $C$, making it non-unique.

## B.1 DIRECT ESTIMATION USING OBSERVATION ONLY

Estimating $z^{(t)}$ through $x^{(t)}$ without considering the history information amounts to maximize the likelihood $p(x^{(t)}|z^{(t)})$ at each time step. Since the observation is a linear mixing of latent variables and Gaussian noise according to equation 11,

$$
x^{(t)}|z^{(t)} \sim N(Cz^{(t)}, R). \tag{12}
$$

Consider all time steps, as long as the $x^{(t)}$ is conditional independent with all other variables given $z^{(t)}$ for all time steps, the conditional distribution:

$$
\begin{aligned}
p(x^{(1:T)}|z^{(1:T)}) &= \prod_{t=1}^{T} N(x^{(t)}; Cz^{(t)}, R) \\
&= \prod_{t=1}^{T} \frac{1}{(2\pi)^{m/2}|R|^{-1}} \exp\left(-\frac{1}{2}(x^{(t)} - Cz^{(t)})^T R^{-1}(x^{(t)} - Cz^{(t)})\right)
\end{aligned}
\tag{13}
$$

Take the log,

$$
\log p(x^{(1:T)}|z^{(1:T)}) = \sum_{i=1}^{T} -\frac{1}{2}(x^{(t)} - Cz^{(t)})^T R^{-1}(x^{(t)} - Cz^{(t)}) - \frac{m}{2}\log 2\pi - \frac{1}{2}\log|R|. \tag{14}
$$

Our goal is to estimate $z^{(t)}$ by maximizing the likelihood above, we further stack the observations and latent variables as:

$$
\mathbf{x} := \begin{bmatrix} x^{(1)} \\ \vdots \\ x^{(t)} \end{bmatrix} \in \mathbb{R}^{mT}, \quad \mathbf{z} := \begin{bmatrix} z^{(1)} \\ \vdots \\ z^{(t)} \end{bmatrix} \in \mathbb{R}^{dT}, \tag{15}
$$

and define the block-diagonal matrices

$$
\mathbf{C}_b := \begin{bmatrix} C & 0 & \dots & 0 \\ 0 & C & \dots & 0 \\ \vdots & \vdots & \ddots & \vdots \\ 0 & 0 & \dots & C \end{bmatrix} \in \mathbb{R}^{mT \times dT}, \quad \mathbf{R}_b = \begin{bmatrix} R & 0 & \dots & 0 \\ 0 & R & \dots & 0 \\ \vdots & \vdots & \ddots & \vdots \\ 0 & 0 & \dots & R \end{bmatrix} \in \mathbb{R}^{mT \times mT}. \tag{16}
$$

Maximizing equation 14 over $\mathbf{z}$ is equivalent to minimizing

$$\mathcal{L}(\mathbf{z}) := \frac{1}{2}(\mathbf{x} - \mathbf{C}_b\mathbf{z})^T\mathbf{R}_b^{-1}(\mathbf{x} - \mathbf{C}_b\mathbf{z}), \tag{17}$$

where all constant terms are dropped. The first order optimality condition is

$$\mathbf{C}_b^T\mathbf{R}_b^{-1}(\mathbf{C_b z} - \mathbf{x}) = \mathbf{0}. \tag{18}$$

Since $\mathrm{rank}(\mathbf{C}_b) \leq mT < dT$, the matrix $\mathbf{B} := \mathbf{C}_b^T\mathbf{R}_b^{-1}\mathbf{C}_b$ is singular and the solution set is an affine subspace; the obtained $\hat{\mathbf{z}}$ is non-unique and ill-posed.

## B.2   INCORPORATE HISTORY VIA A DYNAMIC PRIOR

We note that explicitly consider the contribution from last latent state into the estimation of current state is exactly the Bayes rule that maximize the posterior, whose form is

$$
\begin{aligned}
p(z^{(1:T)}|x^{(1:T)}) &= \frac{p(x^{(1:T)}|z^{(1:T)})p(z^{(1:T)})}{p(x^{(1:T)})} \\
&\propto p(x^{(1:T)}|z^{(1:T)})p(z^{(1:T)}) \\
&= \prod_{t=1}^{T} p(x^{(t)}|z^{(t)}) \prod_{t=2}^{T} p(z^{(t)}|z^{(t-1)})p(z^{(1)}).
\end{aligned}
\tag{19}
$$

Take the log, we have

$$p(z^{(1:T)}|x^{(1:T)}) \propto \sum_{t=1}^{T} \log p(x^{(t)}|z^{(t)}) + \sum_{t=2}^{T} \log p(z^{(t)}|z^{(t-1)}) + \log p(z^{(1)}). \tag{20}$$

In this content, maximizing the posterior above is equivalent to minimize the following log terms where all terms not depending on $z$ are dropped:

- Initial prior

$$\log p(z^{(1)}) := -\frac{1}{2}(z^{(1)} - \mu_0)^T P_0^{-1}(z^{(1)} - \mu_0). \tag{21}$$

- Transition prior

$$\log p(z^{(t)}|z^{(t-1)}) := -\frac{1}{2}(z^{(t)} - Az^{(t-1)})^T Q^{-1}(z^{(t)} - Az^{(t-1)}), \quad t \geq 2. \tag{22}$$

- Likelihood

$$\log p(x^{(t)}|z^{(t)}) := -\frac{1}{2}(x^{(t)} - Cz^{(t)})^T R^{-1}(x^{(t)} - Cz^{(t)}). \tag{23}$$

Similarly, we can stack observations and latents as equation 15, the negative likelihood over multiple steps is

$$
\begin{aligned}
-\log p(x^{(1:T)}|z^{(1:T)}) &= \sum_{t=1}^{T} \log p(x^{(t)}|z^{(t)}) \\
&= \frac{1}{2}(\mathbf{x} - \mathbf{C}_b\mathbf{z})^T\mathbf{R}_b^{-1}(\mathbf{x} - \mathbf{C}_b\mathbf{z}) \\
&= \frac{1}{2}\mathbf{z}^T\mathbf{C}_b^T\mathbf{R}_b^{-1}\mathbf{C}_b\mathbf{z} - \mathbf{x}^T\mathbf{R}_b^{-1}\mathbf{C}_b\mathbf{z} + \text{constant} \\
&:= \frac{1}{2}\mathbf{z}^T\mathbf{B}\mathbf{z} - \mathbf{h}_{\text{data}}^T\mathbf{z} + \text{constant},
\end{aligned}
\tag{24}
$$

where we simplify the notation as $\mathbf{B} := \mathbf{C}_b^T\mathbf{R}_b^{-1}\mathbf{C}_b$, and $\mathbf{h}_{\text{data}} := \mathbf{C}_b^T\mathbf{R}_b^{-1}\mathbf{x}$. Expand the negative transition prior, we have

$$
\begin{aligned}
-\log p(z^{(t)}|z^{(t-1)}) = &\frac{1}{2}z^{(t)T}Q^{-1}z^{(t)} - \frac{1}{2}z^{(t-1)T}A^TQ^{-1}z^{(t)} - \\
&\frac{1}{2}z^{(t)T}Q^{-1}Az^{(t-1)} + \frac{1}{2}z^{(t-1)T}A^TQ^{-1}Az^{(t-1)}.
\end{aligned}
\tag{25}
$$

Expand the negative initial prior

$$-\log p(z^{(1)}) = \frac{1}{2} z^{(1)^T} P_0^{-1} z^{(1)} - z^{(1)^T} P_0^{-1} \mu_0 + \text{constant}. \tag{26}$$

Expand the prior terms over all time steps and collect coefficients of the stacked quadratic $\mathbf{z}^T(\cdot)\mathbf{z}$. The resulting block-tridiagonal weighting matrix is

$$D_{\text{dyn}} = \begin{bmatrix} P_0^{-1} + A^\top Q^{-1} A & -A^\top Q^{-1} \\ -Q^{-1} A & Q^{-1} + A^\top Q^{-1} A & -A^\top Q^{-1} \\ & \ddots & \ddots & \ddots \\ & & -Q^{-1} A & Q^{-1} + A^\top Q^{-1} A & -A^\top Q^{-1} \\ & & & -Q^{-1} A & Q^{-1} \end{bmatrix} \in \mathbb{R}^{Td \times Td}. \tag{27}$$

Similarly, collect coefficients of the stacked linear term $(\cdot)^T \mathbf{z}$. There is no linear term from the dynamics $t = 2, \dots, T$; only the prior mean contributes, in which we define

$$\mathbf{h}_{\text{init}} = \begin{bmatrix} P_0^{-1} \mu_0 \\ 0 \\ \vdots \\ 0 \end{bmatrix}. \tag{28}$$

Now we collect everything, maximizing the posterior in equation 20 is equivalent to minimize

$$\mathcal{J}(\mathbf{z}) := \frac{1}{2} \mathbf{z}^T (\mathbf{B} + \mathbf{D}_{dyn}) \mathbf{z} - (\mathbf{h}_{\text{data}} + \mathbf{h}_{\text{init}})^T \mathbf{z} + \text{constant}. \tag{29}$$

As long as $Q$ and $P_0$ is positive definite, and $\mathbf{B}$ is positive semi-definite, there summation should also be positive definite. Thus, the $\mathcal{J}(\mathbf{z})$ is strictly convex and the solution is unique.

## C  IMPLEMENTATION DETAILS

**Data Preparation.**  For the Sudoku-Extreme training set, we follow the HRM configuration and apply band and digit permutations to each training instance, generating 1,000 augmented variants per example. No data augmentation is used for Maze-Hard. All baselines are trained and evaluated on the same fixed training and test splits.

**Model Architecture.**  To ensure fairness, every baseline uses exactly the same Transformer-layer implementation as its atomic building block: one attention layer, one MLP, and an RMSNorm. The Transformer-layer hyperparameter (e.g., hidden size) and the initialization scheme are kept identical across all baselines.

**Training Protocol.**  All baselines adopt a common backbone and therefore share the same optimization settings, including learning rate, batch size, optimizer, and loss function. Concretely, all experiments use a learning rate of $1 \times 10^{-4}$, a batch size of 768, and the `AdamAtan2` optimizer, with an identical loss function across methods. Each baseline is trained for 60,000 epochs on SUDOKU-EXTREME and, separately, for 60,000 epochs on MAZE-HARD.

**Hardware and Runtime.**  All training runs are executed on $8\times$ AMD MI210 (64 GB) GPUs. The average wall-clock training time is approximately 1 h for SUDOKU-EXTREME and approximately 15 h for MAZE-HARD.

In training and test cost analysis, all measurements were obtained under identical training hyper-parameters using eight AMD MI210 64 GB GPUs. Speeds are reported as batches per second for training and samples per second for inference, with memory averaged per GPU across the eight devices and inference speed computed from total samples divided by wall clock time on the Sudoku Extreme test set.

## D  PSEUDO CODE

```
# ================================
# Hyper-parameters and components
# ================================
m  # number of inner iterations per block (e.g., 16)
n  # number of outer blocks per batch (e.g., 16)

T     # the ONLY recurrent transformer layer
Head  # readout head producing y from z
Opt   # optimizer over parameters of T and Head
Loss  # criterion, e.g., cross-entropy / L2

def init_state(batch):
    """z0 = 0 for each batch."""
    return zeros_like_state(batch)

# ================================
# Training loop
# ================================
for epoch in range(num_epochs):
    for batch in DataLoader:
        x, target = batch
        z = init_state(batch_size(x))   # z0 = 0

        # ---- Periodic Alignment  ----
        for block_idx in range(1, n + 1):

            Opt.zero_grad(set_to_none=True)   # we update once per block

            # ---- Flat recurrent Transformer ----
            for t in range(1, m + 1):

                if block_idx == 1:
                    # Only in the 1st block do reflection injection of x
                    # Inject both x and the previous z into next z.
                    # ---- Reflective Represent Learning ----
                    z_in = ReflectFuse(z, x)
                else:
                    # From the 2nd block on, DO NOT inject x anymore
                    z_in = z                        # use z only

                # One layer applied repeatedly (weight sharing)
                z = T(z_in)                         # same layer, different
                    time step

            # ---- End of a block: produce Y and train immediately ----
            y_pred = Head(z)                        # predict from current z

            cur_target = target if is_scalar_label(target) else
                target[block_idx]

            loss = Loss(y_pred, cur_target)
            # backprop within this block only
            loss.backward()
            Opt.step()                              # update T and Head now

            # ---- Detach for the next block ----
            # Break the graph so gradients from the next block cannot
            # flow back into the current/previous blocks.
            z = z.detach()
```

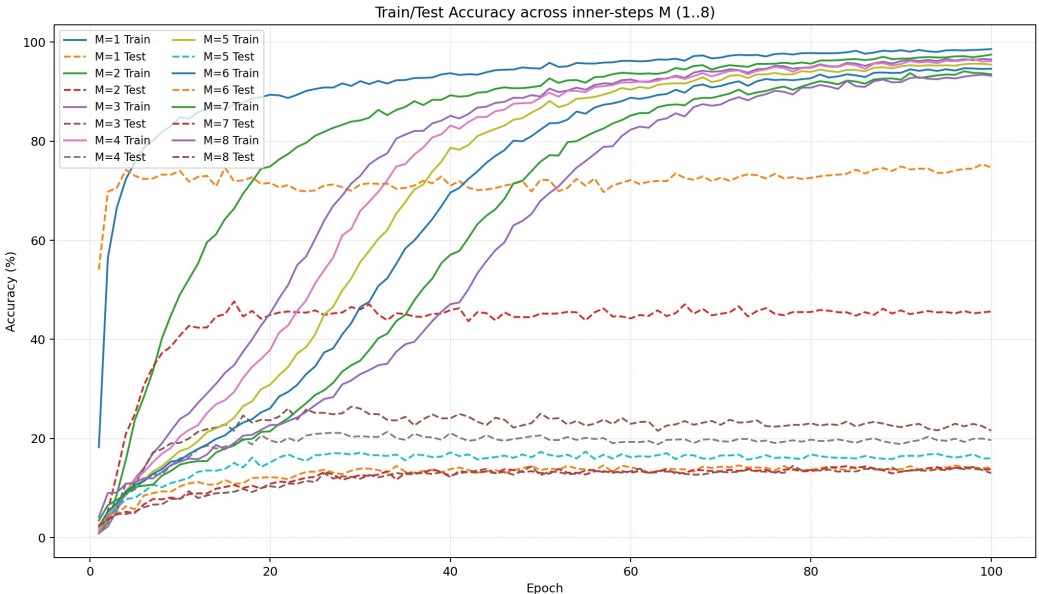

Figure 7: Analysis of the applicability of $SR^2$. M denotes how many times we forward the model within a single alignment (supervision) step.

We provide the pseudo code for **$SR^2$**. This pseudo code serves as a high-level description of the algorithm's workflow, outlining its core computational steps and logical structure.

# E  ANALYSIS OF $SR^2$ APPLICABILITY

From another perspective, not all tasks require iterative updates. For relatively simple problems such as image classification where the mapping from input to output is more direct and the latent space is less entangled, a single forward pass is often sufficient. To probe this, we instantiated the $SR^2$ architecture on CUB-200-2011 (Caltech–UCSD Birds 200–2011) Wah et al. (2011), a fine-grained classification benchmark, using the same backbone and comparable parameter budget as a standard ViT baseline, identical training data splits, and a conventional top-1 accuracy protocol. As summarized in Fig. 7, the ViT baseline attains roughly 75% top-1 accuracy, whereas $SR^2$ equipped with *Reflection* and *Self-Refinement* reaches under 45%. This gap, together with the learning curves in the figure, supports our claim that iterative refinement is especially advantageous in complex constraint-satisfaction or reasoning settings, but offers limited benefits and can even be detrimental when the latent structure is comparatively simple and direct, as in standard classification.

## DISCLOSURE OF LLM USAGE

In accordance with the ICLR 2026 policy on the use of Large Language Models (LLMs), we disclose that LLMs were used solely to assist with grammar refinement and language polishing in this paper. No part of the research ideation, experimental design, implementation, or analysis relied on LLMs. The authors take full responsibility for the technical content and conclusions of the work.

