# OpenReview forum: "Selection, Reflection and Self-Refinement: Revisit Reasoning Tasks via a Causal Lens"
_ICLR.cc/2026/Conference — ICLR 2026 Poster_

### Official Review · Reviewer_nGiF · 2025-10-21

**Soundness:** 2
**Presentation:** 2
**Contribution:** 2
**Rating:** 4
**Confidence:** 3

**Summary:**

This paper formalize reasoning problem as selection mechanisms through a causal perspecitive. Based on this formulization, the paper summarizes two hypothesis in reasoning tasks, which inspires the SR2 framework. The framework define reflection as a fixed-point problem and used a flatten recurrent transformer to solve it. Further, the dependency self-refinement and periodic alignment are proposed to learn dependency and formalize training.

**Strengths:**

1. The idea of formalizing reflection as a fixed-point equation is novel and interesting.
2. The proposed method achieved good performance on Sudoku-Extreme and Maze-Hard dataset with less number of parameters.

**Weaknesses:**

1. The paper is a bit hard to follow (See question 1-2).
2. A general performance comparison with baselines are missing, including training / test cost.
3. Adding more benchmarks (e.g. ARC-AGI) would be helpful. One of my concern on proposed framework is the fixed-point formulation may only works for problems with specific structure, not all reasoning tasks.

**Questions:**

1. In Ln. 162 - 169, is there a formal representation of $\mathcal{S}$ or $z$? E.g. How is $z$ represented by $x, y$? I think as an example it should illustrate how to formalize rules in reasoning problems into selection variables $z$ and constraints $S$, instead of leaving an abstract expression as in Eq. (2).
2. What is the motivation of formalizing $z$ as the solution a fixed-point equation (3)? How is this related to hypothesis 1? The connection is unclear.

---

> ### Author Response · Authors · 2025-11-21
> **Response to Reviewer nGiF**
>
> Dear Reviewer nGiF,
>
> We thank you for your insightful review, for recognizing the novelty of our fixed‑point formulation of reflection, and for your positive assessment of the empirical performance of our framework. Your feedback on clarity, benchmarking, and the generality of our approach has been very helpful in strengthening the presentation. In response, we have added a detailed comparison of training and inference cost, extended our experiments to the ARC‑AGI benchmark, provided a concrete example to illustrate how X, Y, z, and S are instantiated, and clarified the motivation for the fixed‑point formulation and its connection to Hypothesis 1. We trust that these revisions make the paper clearer and more complete; the corresponding point‑by‑point responses are given below.
>
>
> > **W2:** A general performance comparison with baselines are missing, including training / test cost.
>
>
> Thank you for your constructive suggestion which helps us present the efficiency of SR² more comprehensively. Prompted by your comment, we measured SR²’s speed during both training and inference and compared it against the baselines. As shown in the table below, the Periodic Alignment module makes SR² slightly slower than Direct Predict in both training and inference. However, because SR² uses only a single layer with no hierarchical structure, it is faster than HRM, which relies on a multi-level hierarchical design. In terms of training memory cost, SR² consumes slightly more GPU memory than the other baselines. We have incorporated this analysis of training and inference cost into Section 3.4 of the revised manuscript.
>
> |      Method       | Training Speed (Batch/s)$^{1}$ | Training Memory (GB)$^{2}$ | Inference Speed (Sample/s)$^{3}$ |
> | ----------- | ------------------------------ | -------------------------- | -------------------------------- |
> | Direct Pred | 21.39                          | 3.024                      | 7489.6                           |
> | HRM         | 10.57                          | 3.231                      | 1487.7                           |
> | SR²         | 14.73                          | 3.950                      | 2073.6                           |
>
> $^{1}$ We evaluate the training cost of all baselines on the Sudoku-Extreme training set using 8× AMD MI210 64GB GPUs. For every baseline, we use the same batch size and learning rate; training speed is reported in batches per second (higher is faster).
>
> $^{2}$ For training GPU memory, we report the average memory usage per GPU, averaged over all 8 GPUs.
>
> $^{3}$ We measure inference speed on the Sudoku-Extreme test set using 8× AMD MI210 64GB GPUs. We record the total wall-clock time to complete inference on the full test set, and compute speed as total number of samples divided by runtime (samples per second; higher is faster).
>
>
>
> > **W3:** Adding more benchmarks (e.g. ARC-AGI) would be helpful. One of my concern on proposed framework is the fixed-point formulation may only works for problems with specific structure, not all reasoning tasks.
>
> Thank you for encouraging us to broaden our empirical evaluation. In response, we have evaluated SR² and the baselines on the ARC-AGI benchmark under identical settings. We trained all methods on 8×H200 NVL and summarize the results in the table below. The results show that SR² achieves consistent improvements on both ARC-1 and ARC-2 while using only one-eighth of the parameters of the baselines, which helps alleviate the concern that the fixed-point–style formulation would only work for a very narrow class of architectures. The above experimental results have been added to Table 1 in the revised version.
>
>
>
> | Method         | Params | ARC-1 ${^1}$ | ARC-2 |
> | -------------- | ------ | ------ | ----- |
> | Direct Predict | 27.3M  | 21.0   | 0.0   |
> | HRM            | 27.3M  | 40.3   | 5.0   |
> | SR²            | 4.3M   | 44.3   | 6.7   |
>
>
> We note that training on ARC-AGI requires substantial compute (e.g., around 24 hours on 8×H100), which exceeded our original budget and prevented us from running this experiment before submission. After receiving your feedback, we secured additional resources and then conducted the ARC-AGI experiments described above. We hope that such experiments can solve your concerns well.
>
> ${^1}$ We follow the official ARC-AGI evaluation protocol and report pass@2 accuracy (%) on both ARC-1 and ARC-2.

---

> > ### Author Response · Authors · 2025-11-21
> > **Response to Reviewer nGiF (Part II)**
> >
> > > **Q1:** In Ln. 162 - 169, is there a formal representation S of or z? E.g. How is z represented by x,y ? I think as an example it should illustrate how to formalize rules in reasoning problems into selection variables z and constraints S, instead of leaving an abstract expression as in Eq. (2).
> >
> >
> >
> > Thanks! Your suggestion to include a concrete example indeed helps clarify how our method is instantiated. In the revised manuscript Section 2.1, we have added an explicit example that illustrates the concrete meanings of $X$, $Y$, $z$, and $S$. Using Figure 1 as an example, $S$ denotes the selection mechanism, and $z$ denotes a specific rule that characterizes the relationship between $X$ and $Y$. The Boolean value $S(z)$ (or $S_z$) indicates whether the current pair $(X, Y)$ satisfies rule $z$.
> >
> > Concretely, consider the purple row in Figure 1. Let $Y_{ij}$ be the number to be filled in the $i$-th row and $j$-th column, and let $X_{\text{in-row}}$ denote all already-filled numbers in the same row. If $Y_{ij}$ is different from every number in $X_{\text{in-row}}$, we set $S^i_{\text{row}}=1$; otherwise, we set $S^i_{\text{row}} = 0$. Here, $z = \text{row}$ encodes the rule that each row must contain every digit exactly once which means the digits in $X_{\text{in-row}}$ together with $Y_{ij}$ must all be distinct.
> >
> >
> >
> > > **Q2:** What is the motivation of formalizing z as the solution a fixed-point equation (3)? How is this related to hypothesis 1? The connection is unclear.
> >
> >
> >
> > We appreciate this question, as it gives us the opportunity to clarify the motivation to design Eq. (3). Our motivation is to introduce a reflection mechanism whose core idea is that, when tackling a complex reasoning problem, the model should not produce the solution $z$ in a single step. Instead, it should iteratively refine $z$ through trial-and-error to progressively approach the ground-truth solution. This is conceptually related to Chain-of-Thought reasoning in LLMs, but in our case the iterative reasoning process is represented by an implicit latent variable rather than explicit token sequences.
> >
> > In Hypothesis 1, we posit that for reasoning tasks the latent space is much larger than the observation space. Because this latent space is so large, it is difficult to directly predict a fully correct latent variable $z$ from the input $x$ in one shot. The fixed-point formulation and iterative refinement are introduced precisely to address this challenge: instead of attempting to obtain $z$ in a single forward pass, we repeatedly update the current estimate of $z$ during the reasoning process so that it moves closer and closer to the ground truth.
> >
> > For example, in a Sudoku task, it is hard to correctly fill in all missing digits at once. However, it is much more feasible to fill in one (or a few) digits at a time, check whether the partially completed grid satisfies the constraints, and then continue filling based on the current state. This iterative “write–check–refine” process significantly improves the probability of arriving at a correct full solution.
> >
> > We have updated the discussion around Eq. (3) in Section 2.3 of the revised manuscript to make this motivation explicit and to better connect the fixed-point view of $z$ with the intended reflection mechanism.

---

> > > ### Comment · Reviewer_nGiF · 2025-11-25
> > >
> > > Thank you for the detailed response! I'll update my score to 6. One minor typo: In Ln. 173 of the revised draft, there should be a period after "Figure 1b"?

---

> > > > ### Author Response · Authors · 2025-11-26
> > > > **Thank you for your comments and raised score; we are pleased your concerns are well addressed.**
> > > >
> > > > Thank you very much for your careful reading and for pointing out the typo at Line 173. We appreciate your attention to detail. We have corrected the missing period after “Figure 1b” in the revised version. Thank you again for your thoughtful feedback and for updating your score.

---

### Official Review · Reviewer_v12F · 2025-10-26

**Soundness:** 3
**Presentation:** 2
**Contribution:** 2
**Rating:** 6
**Confidence:** 4

**Summary:**

This paper frames reasoning tasks as causal selection mechanisms, where latent reasoning rules constrain observed space. The authors propose two hypotheses including 1/ latent space complexity exceeding observations and 2/ dense interdependencies among latents. Based on these two hypotheses, the authors propose the SR2 framework to address the challenges involved in reasoning. Specifically, SR2 integrates a reflective representation learning module, a dependency self-refinement module, and a periodic alignment module. Evaluations on Sudoku and Maze show >10% accuracy gains over Hierarchical Reasoning Model (Wang el al., 2025) with 8x fewer parameters.

**Strengths:**

1. The causal lens provides a structured view of reasoning difficulties, potentially useful for latent modeling in ML.
2. Hypotheses on latent complexity and dependencies are insightful and well-illustrated with examples like Sudoku.
3. Empirical gains in reasoning efficiency when model size is small.

**Weaknesses:**

1. As acknowledged by authors, benchmarking of this paper is limited to Sudoku and Maze Navigation with clear and simple logical constraints. This may limit the claims of generality. Suggesting adding additional evaluations on STEM tasks.
2. Missing discussion of related work. The reflection framework of SR2 is related to the established line of work on self-correction techniques. For example, the Reflexion paper (Shinn el al., 2023) introduced verbal self-reflection.

**Questions:**

The introduction mentioned scaling-based progress (e.g., Kaplan et al., 2020). With your efficiency gains, do you foresee SR2 scaling to frontier LLMs, or is it better suited for smaller, specialized models?

---

> ### Author Response · Authors · 2025-11-21
> **Response to Reviewer v12F (Part I)**
>
> Dear Reviewer v12F,
>
> We are grateful for your careful review and for your positive remarks on our causal framing of reasoning, the hypotheses on latent complexity and dependencies, and the empirical gains in reasoning efficiency. Your suggestions regarding benchmark coverage, the connection to self‑correction methods such as Reflexion, and the scalability of SR² to larger LLMs have been instrumental in improving the manuscript. In the revision, we have broadened our empirical evaluation, expanded the related‑work discussion on self‑correction, and clarified the applicability of SR² to larger models. We hope these changes resolve the issues you raised; further details are provided in the point‑by‑point responses below.
>
> > **W1:** As acknowledged by authors, benchmarking of this paper is limited to Sudoku and Maze Navigation with clear and simple logical constraints. This may limit the claims of generality. Suggesting adding additional evaluations on STEM tasks.
>
> Thank you for the constructive suggestion. Following your advice, we expanded our evaluation beyond Sudoku and Maze Navigation by adding experiments on the ARC‑AGI benchmark [1]. Under the same experimental setting as HRM, we report side‑by‑side results for SR^2, a standard Transformer baseline, and a recent strong baseline, HRM. The results (included in Table 1 of the revised manuscript) show that even on this broader and more challenging benchmark, SR^2 achieves consistent improvements while using only 1/8 of the parameters.
>
> | Method         |  Params  |ARC-1$^{1}$  | ARC-2 |
> | -------------- |  -----   | ----- | ----- |
> | Direct Predict |  27.3M   | 21.0  | 0.0   |
> | HRM            |  27.3M   | 40.3  | 5.0   |
> | SR²            |  4.3M    | 44.3  | 6.7   |
>
>
> $^{1}$ We follow the official ARC-AGI evaluation protocol and report pass@2 accuracy (%) on both ARC-1 and ARC-2.
>
>
>
> > **W2:** Missing discussion of related work. The reflection framework of SR2 is related to the established line of work on self-correction techniques. For example, the Reflexion paper (Shinn el al., 2023) introduced verbal self-reflection.
>
> Thanks! This comment is very helpful for situating our work in the broader literature. In response, we have added a discussion of LLM self-correction techniques to Appendix A of the revised manuscript. In brief, a growing body of work investigates explicit self-correction for LLMs via verbal feedback, critique, and planning. Reflexion [2] introduces verbal self-reflection and episodic memory to iteratively improve task success; Self-Refine [3] performs critique-and-edit loops using model-generated feedback; Self-Consistency [4] reduces reasoning errors by sampling and aggregating multiple chains of thought; STaR [5] bootstraps rationales through iterative data augmentation and fine-tuning; and Constitutional AI [6] uses a set of principles to guide model-written critiques and refine outputs.
>
> While related in spirit, our approach differs in that SR² does not explicitly generate token sequences to steer self-correction. Instead, the correction process is implemented implicitly through iterative updates of the latent variables (z), integrating self-correction into the latent reasoning dynamics without additional verbal overhead.
>
>
> References
> ---
> [1] [https://arcprize.org/arc-agi](https://arcprize.org/arc-agi)
>
> [2] Shinn, N., Cassano, F., Berman, E., Gopinath, A., Narasimhan, K., & Yao, S. (2023). *Reflexion: Language agents with verbal reinforcement learning*. arXiv preprint arXiv:2303.11366.
>
> [3] Madaan, A., Tandon, N., Gupta, P., Hallinan, S., Gao, L., Wiegreffe, S., Alon, U., Dziri, N., Prabhumoye, S., Yang, Y., Gupta, S., Majumder, B. P., Hermann, K., Welleck, S., Yazdanbakhsh, A., & Clark, P. (2023). *Self-Refine: Iterative refinement with self-feedback*. arXiv preprint arXiv:2303.17651.
>
> [4] Wang, X., Wei, J., Schuurmans, D., Le, Q., Chi, E., Narang, S., Chowdhery, A., & Zhou, D. (2022). *Self-consistency improves chain of thought reasoning in language models*. arXiv preprint arXiv:2203.11171.
>
> [5] Zelikman, E., Wu, Y., Mu, J., & Goodman, N. D. (2022). *STaR: Bootstrapping reasoning with reasoning*. arXiv preprint arXiv:2203.14465.
>
> [6] Bai, Y., Kadavath, S., Kundu, S., Askell, A., Kernion, J., Jones, A., … Kaplan, J. (2022). *Constitutional AI: Harmlessness from AI feedback*. arXiv preprint arXiv:2212.08073.

---

> > ### Author Response · Authors · 2025-11-21
> > **Response to Reviewer v12F (Part II)**
> >
> > > **Q1:** The introduction mentioned scaling-based progress (e.g., Kaplan et al., 2020). With your efficiency gains, do you foresee SR2 scaling to frontier LLMs, or is it better suited for smaller, specialized models?
> >
> >
> > Thank you for the question about how SR² relates to scale-driven progress in LLMs. Conceptually, SR² is designed to trade additional test-time compute for deeper latent reasoning, making it naturally compatible with frontier-scale models rather than confined to small, specialized architectures.
> >
> > Given current compute constraints, we have not yet pretrained an LLM with the SR² architecture from scratch. That said, another recent study suggest that SR²-style approaches hold substantial promise for text-based reasoning in LLMs. Several weeks ago, a study from ByteDance Seed integrated a module analogous to SR²’s self-refinement component into an LLM and reported strong reasoning performance; unlike SR², their method did not employ Reflective Representation Learning or Periodic Alignment. We view this as a potential evidence of headroom to scale SR² to larger parameter regimes. In future work, we will scale up SR² to further explore its capabilities as soon as we obtain more computational resources.

---

### Official Review · Reviewer_o7wn · 2025-10-31

**Soundness:** 2
**Presentation:** 2
**Contribution:** 3
**Rating:** 4
**Confidence:** 4

**Summary:**

This paper interprets reasoning tasks such as Sudoku and maze solving as modeling problems governed by latent variables that act as selection constraints. It argues that the latent space underlying reasoning exhibits high complexity and strong interdependencies among variables, which explains why existing models still struggle to reason effectively even after large-scale training. This paper proposes a new reasoning framework SR^2, designed to model these latent dependencies and iteratively refine latent variables. SR^2 outperforms recent baselines such as the Hierarchical Reasoning Model (HRM), achieving higher performance with fewer parameters, which supports the motivation and design of the proposed framework.

**Strengths:**

The proposed framework redefines reasoning as a latent-variable-based constraint mechanism, explaining the difficulty of reasoning tasks through the perspective of latent complexity and dense dependencies.

The SR^2 architecture consists of selection, reflection, and self-refinement modules, which directly models and optimizes latent variables, achieving improved performance on Sudoku and Maze tasks with a smaller parameter budget.

**Weaknesses:**

The connection between Eq. 1 and the proposed framework remains unclear. Eq. 1 defines a generative process from latent rule variables z to observed pairs (x, y) and introduces the notion of selection constraints S derived from z. However, the way these elements are instantiated or represented within SR^2 is not clearly explained.

The experiments do not include the ARC task, leaving the reasoning ability insufficiently validated.

This paper lacks discussion of the training cost. If supervision is applied to different hierarchical levels of latent variables z, the model may face risks of gradient instability, potentially increasing computational overhead.

**Questions:**

How does the framework model the process described by Eq. 1? e.g., how are the rule variables z represented, and how do they enforce corresponding constraints within SR^2? Which part of SR^2 corresponds to the function g in Eq. 1?

Are there difficults that prevent SR^2 from handling ARC-style reasoning tasks?

The paper claims that SR^2 “explicitly integrates Selection, Reflection, and Self-Refinement.” How is the Selection component concretely implemented in the architecture?

Compared to previous SOTA methods, how does the training cost of SR^2 (in time or memory) scale?

---

> ### Author Response · Authors · 2025-11-21
> **Response to Reviewer o7wn (Part I)**
>
> Dear Reviewer o7wn,
>
> We appreciate your thoughtful review and the constructive feedback you provided on both the theoretical formulation and the empirical evaluation. We are glad that you found our latent‑variable view of reasoning and the SR² architecture to be of interest. In response to your comments, we have clarified how the theoretical formulation in Eq. (1) is instantiated in SR², added ARC‑AGI experiments to further validate its reasoning ability, and expanded our discussion of how Selection is implemented as well as how the training cost and stability of SR² compare with prior methods. We believe these revisions directly address your concerns, and we detail them in the following point‑by‑point responses.
>
>
> > **W1:** The connection between Eq. 1 and the proposed framework remains unclear. Eq. 1 defines a generative process from latent rule variables z to observed pairs (x, y) and introduces the notion of selection constraints S derived from z. However, the way these elements are instantiated or represented within SR^2 is not clearly explained.
>
> > **Q1:** How does the framework model the process described by Eq. 1? e.g., how are the rule variables z represented, and how do they enforce corresponding constraints within SR^2? Which part of SR^2 corresponds to the function g in Eq. 1?
>
> Thank you for the question which helps clarify the relationship between the equation in our paper and the model architecture. In fact, Eq. (1) describes the data‑generating process for our task, rather than the design of the model itself. Specifying a clear data‑generating process provides a foundational basis for causal analysis (e.g., [1]).
>
>
> In the implementations, we try to learn the conditional distribution $p(y \mid x)$, instead of the joint distribution $p(x, y)$ (the data generation process) on the left‑hand side of Eq. (1). The conditional distribution $p(y \mid x)$ can be easily derived from the joint distribution, so that we can directly align this target with our implementations. Specifically,
>
> $$
> p(x,y) = \int p(z) p_g(x,y\mid z) \mathbf{1}\[S(z)=1\] dz \qquad \text{(1)}
> $$
>
> From this, we obtain
>
> $$
> p(y\mid x) = \int p(z\mid x) p_g(y\mid x,z) \mathbf{1}[S(z)=1] dz
> $$
>
>
> Where:
>
> - $p(y \mid x)$ is the overall learning objective: given input $x$, predict the distribution of outputs $y$.
> - $z$ is a latent variable corresponding to the hidden state produced by the Transformer layer in our architecture.
> - $p(z \mid x)$ is implemented by the Transformer layer as a function z = f(x): it takes $x$ as input and predicts the associated latent variable $z$. Furthermore, to do the reasoning, we propose the relection model, which replaces the naive function z=f(x) to z = f(x,z). In such case, both inputs x and the previous state of z are the inputs of the Transformer.
> - Given our causal modeling, since the latent variable $z$ already encodes information about $x$, the final `lm_head` directly learns $p(y \mid z)$ as a surrogate for $p(y \mid x, z)$ during training, mapping latent variables to logits.
>
> This transformation makes explicit how the data‑generating process connects to our model. We have added a more detailed explanation of Eq. (1) to Section 2.1 in the revised version.
>
>
> > **W2:** The experiments do not include the ARC task, leaving the reasoning ability insufficiently validated.
>
> > **Q2:** Are there difficults that prevent SR^2 from handling ARC‑style reasoning tasks?
>
> Thanks for your valuable suggestions which had improved the completeness of our experiments and justifications. In light of your suggestion, we obtained additional compute budget, prioritized this experiment, and trained SR² on 8×H200 NVL using exactly the same setup as the reported baselines. The new results, now included in the revised paper, show that SR² consistently outperforms both a standard Transformer (“Direct Predict”) and a strong hierarchical baseline (HRM), while using only about one-eighth of their parameters:
>
>
> | Method         |  Params  |ARC-1¹  | ARC-2 |
> | -------------- |  -----   | ----- | ----- |
> | Direct Predict |  27.3M   | 21.0  | 0.0   |
> | HRM            |  27.3M   | 40.3  | 5.0   |
> | $\text{SR}^{2}$            |  4.3M    | 44.3  | 6.7   |
>
>
> ¹ We follow the official ARC-AGI evaluation protocol and report pass@2 accuracy (%) on both ARC-1 and ARC-2.
>
> We would also like to clarify why these results were not included at submission time. Training on ARC-AGI requires substantial computation (e.g., roughly 24 hours on 8×H100 GPUs), which exceeded our original compute budget, so we had not run this experiment before the initial submission. After receiving your review, we secured additional resources and were able to carry out the ARC-AGI experiments as described above. These ARC-1 and ARC-2 benchmark results have also been incorporated into Table 1 of the revised manuscript.

---

> > ### Author Response · Authors · 2025-11-21
> > **Response to Reviewer o7wn (Part II)**
> >
> > > **Q3:** The paper claims that SR^2 “explicitly integrates Selection, Reflection, and Self‑Refinement.” How is the Selection component concretely implemented in the architecture?
> >
> >
> > We are grateful for this question, as it helps us clarify the connection between the SR² theory and the concrete architecture. In light of your suggestion, we have refined the corresponding sentence in Section 2.3 to make this clearer:
> > “SR² implements the Selection Mechanism via the Reflection and Self-Refinement modules.”
> >
> > At the conceptual level, Selection Mechanism is the organizing principle that governs the entire SR² architecture, rather than just a standalone module within the framework. In our concrete implementation, the global training objective is to learn a selection mechanism $S$ that, given an input $X$, identifies the unique pair $(X, Y)$ that satisfies a set of task-specific constraints, where $Y$ is the model’s output. In this sense, the Selection mechanism corresponds to the overall SR² model, while the Reflection and Self-Refinement components are two concrete modules through which this mechanism is instantiated and optimized.
> >
> > > **W3:** This paper lacks discussion of the training cost. If supervision is applied to different hierarchical levels of latent variables z, the model may face risks of gradient instability, potentially increasing computational overhead.
> > > **Q4:** Compared to previous SOTA methods, how does the training cost of SR^2 (in time or memory) scale?
> >
> > This is an important point, and we fully agree that comparing training cost is essential for a fair assessment. In the revised manuscript, Section 3.4 now includes a detailed cost comparison against our main baselines. Representative results show that SR² trains faster than the hierarchical baseline HRM. In terms of memory, the smaller parameter count of SR² reduces the memory needed for model states; however, due to our recurrent design with Periodic Alignment, each step optimizes gradients for $M$ “copies” of the model, so the gradient-storage component of memory is higher than the baselines. The overall training speed and memory comparison is summarized in the table below:
> >
> >
> > |                | Training Speed (Batch/s)$^{1}$ | Training Memory (GB)$^{2}$ |
> > | -------------- | --------------------------- | ----------------------- |
> > | Direct Predict | 21.39                       | 3.024                   |
> > | HRM            | 10.57                       | 3.231                   |
> > | SR²            | 14.73                       | 3.950                   |
> >
> >
> > Regarding the risk of gradient instability: SR² uses a **single** latent variable (z), rather than multiple hierarchical latent levels. As a result, we do not apply supervision across different latent tiers, and we therefore do not encounter the type of instability raised in your comment.
> >
> > $^{1}$ All training-cost measurements are obtained on 8× AMD MI210 64 GB GPUs. For every baseline, we use the same batch size and learning rate; training speed is reported as batches per second (higher is faster).
> >
> > $^{2}$ For GPU memory comparison, we report the average training-time memory usage per GPU, averaged over all 8 GPUs.
> >
> >
> > ---
> > References
> >
> > [1] Yao, D., Xu, D., Lachapelle, S., Magliacane, S., Taslakian, P., Martius, G., von Kügelgen, J., & Locatello, F. (2024). *Multi-view causal representation learning with partial observability.* In The Twelfth International Conference on Learning Representations (ICLR 2024 – Spotlight).

---

### Official Review · Reviewer_qWVm · 2025-10-31

**Soundness:** 4
**Presentation:** 4
**Contribution:** 3
**Rating:** 8
**Confidence:** 3

**Summary:**

This paper proposes a novel causal formulation of reasoning tasks, framing them as selection mechanisms in which logical concepts act as operators that constrain observed inputs. Building on this, the authors identify two key hypotheses—latent space complexity (the reasoning latent space is exponentially larger than the observation space) and dense interdependence (latent variables are strongly coupled)—as root causes of reasoning difficulty. To address these challenges, the authors introduce SR² (Selection, Reflection, and Self-Refinement), a framework that explicitly models iterative latent-space refinement. SR² includes three modules: (1) reflective representation learning that iteratively updates latent variables conditioned on inputs, (2) dependency self-refinement that enforces latent consistency by removing input signals, and (3) periodic alignment to stabilize training. The approach uses a flattened recurrent Transformer architecture and is evaluated on Sudoku-Extreme and Maze-Hard reasoning benchmarks, achieving up to 10% higher accuracy with 8× fewer parameters than prior state-of-the-art models (e.g., HRM).

**Strengths:**

* Novel causal perspective: Formulating reasoning as a selection mechanism is a theoretically original angle connecting causality and reasoning.
* Well-structured SR² pipeline: The three-module design (reflection, self-refinement, periodic alignment) is coherent and practically motivated by the stated hypotheses.
* Strong empirical results: Outperforms HRM (ICLR 2025) by 11.6% on Sudoku-Extreme and 19.2% on Maze-Hard, showing competitive reasoning ability with much smaller models.

**Weaknesses:**

* Limited empirical scope: Evaluations on Sudoku and Maze tasks, while useful, do not establish general reasoning improvement beyond symbolic settings. It would be interested to see if the approach can be applied to more open-ended domains such as math problem with natural language as the interface. How SR² behaves under large-scale LLM reasoning (e.g., text-based chain-of-thought) is not explored.
* Ablations vs. causality: While the causal “selection mechanism” framing is conceptually appealing, the improvements could be fully explained by iterative refinement and alignment, independent to the causal lens.

**Questions:**

Q1: Authors state that the latent space is “exponentially larger” than the observation space. Is this a theoretical claim that can be formalized (e.g., by counting admissible configurations in Sudoku), or an empirical observation? How could one measure latent complexity in your setup?

Q2: The system is majorly motivated from selection mechanism, however the system is more inherited from DEQ or Implicit Deep Learning frameworks. How does SR² differ theoretically from them? Is there any convergence guarantee for your fixed-point iterations?

---

> ### Author Response · Authors · 2025-11-21
> **Response to Reviewer qWVm (Part I)**
>
> Dear Reviewer qWVm,
>
> Thank you very much for your careful reading of our manuscript and for your highly encouraging overall assessment. We are especially grateful for your insightful comments on the empirical scope, causal framing, latent‑space complexity, and the relation to DEQ/implicit models, which have significantly helped us refine both the conceptual and empirical aspects of the work. In the revised version, we clarify the role of the selection‑mechanism perspective, formalize our discussion of the latent space, and add an explicit theoretical and empirical comparison to DEQ, including a brief convergence discussion. We deeply appreciate the time and thought you invested in our paper, and we have incorporated your feedback in detail in the point‑by‑point responses below.
>
> > **W1:** Limited empirical scope: Evaluations on Sudoku and Maze tasks, while useful, do not establish general reasoning improvement beyond symbolic settings. It would be interested to see if the approach can be applied to more open-ended domains such as math problem with natural language as the interface. How SR² behaves under large-scale LLM reasoning (e.g., text-based chain-of-thought) is not explored.
>
> Thanks a lot for your suggestion. We totally agree that exploring the generalization of the SR² framework to natural‑language reasoning tasks is both promising and important. At present, our computational budget does not allow us to pretrain an SR²‑based LLM from scratch. Nevertheless, we note that a paper from ByteDance Seed published about three weeks ago [1] reports a related attempt in LLM reasoning: they adopt a design similar to our Self‑Refinement module to build LoopLM (Loop Language Models) for latent reasoning. Unlike our approach, their model does not incorporate a Reflection module, nor do they employ Periodic Alignment during training. We view this result as preliminary evidence that SR²‑style frameworks have substantial potential for LLM reasoning. Accordingly, we will prioritize extending SR² to more modalities, especially text‑based reasoning in our future work.
>
>
> > **W2:** Ablations vs. causality: While the causal “selection mechanism” framing is conceptually appealing, the improvements could be fully explained by iterative refinement and alignment, independent to the causal lens.
>
> Thank you for your valuable question. We fully agree that, at the implementation level, most of the performance gains come from the technical implementations such as iterative refinement and alignment procedure. We would like to highlight that these implementations are directly motivated by the underlying selection mechanism; the mechanism and its implementation are not two separate modules, but within the relation of formulation and implementation. By this formulation, we aim to provide more insights on why such iterative refinement and alignment implementations are necessary for a reasoning task, so that readers can better understand under which conditions this refinement process should or should not be used.
>
> To this end, we provide one possible explanation from a causal perspective. Specifically, we hypothesize that the latent space involved in reasoning tasks is highly complex and strongly dependent (Hypothesis 1 and Hypothesis 2).
>
> For reasoning tasks, we argue that both the observed problem and the desired answer must satisfy certain constraints, or equivalently, a selection mechanism. Solving such tasks can therefore be viewed as a constraint satisfaction problem. The main difficulty lies in inferring the true underlying constraints from only a few examples, since there may be many seemingly plausible candidates. For instance, in a Sudoku puzzle, there are multiple interlocking rules (e.g., each row, column, and subgrid must contain all digits exactly once), and these rules are interdependent. This interdependence effectively forces the model to perform iterative updates and alignment in order to converge to a globally consistent solution.
>
> From another perspective, not all tasks require iterative updates. For relatively simple problems such as classification, where the mapping from input to output is more direct and the latent space is less entangled, a single forward pass is often sufficient. We have tried applying the SR² architecture to CUB-200-2011 (Caltech-UCSD Birds 200-2011)[2], which is a fine-grained bird image classification benchmark. However, the results show that, compared with a standard ViT baseline that reaches roughly 75% performance, the SR² method equipped with Reflection and Self‑Refinement achieves less than 45%. From this observation, we infer that iterative refinement offers greater advantages in more complex reasoning scenarios than in relatively simple prediction tasks. We have added the relevant experimental details in Appendix E of the revised version.

---

> > ### Author Response · Authors · 2025-11-21
> > **Response to Reviewer qWVm (Part II)**
> >
> > > **Q1:** Authors state that the latent space is “exponentially larger” than the observation space. Is this a theoretical claim that can be formalized (e.g., by counting admissible configurations in Sudoku), or an empirical observation? How could one measure latent complexity in your setup?
> >
> > We appreciate the opportunity to make our statement about the relative sizes of the latent and observation spaces more precise. In the setting of our paper, the claim that “the latent space is exponentially larger than the observation space” can in fact be formalized as a theoretical statement.
> >
> > Using the Sudoku task as an example, we understand the latent space as a rule space, which can be parameterized as the set of all possible valid solution trajectories. Each trajectory is defined as follows: starting from the initial grid, at each step we (i) choose one still-empty cell according to some ordering, and (ii) fill it with the correct digit from a full solution, while ensuring that all row/column/subgrid constraints are satisfied at every step. By contrast, the observation space consists only of the digits that are already revealed in the partially filled Sudoku grid.
> >
> > For the instance shown in Figure 1, suppose we mask out $n$ cells and denote by $S$ the number of complete solutions to that puzzle (in our dataset we guarantee (S = 1)). Then the size of the latent space can be approximated as
> > $$
> > \boxed{|\mathcal{L}| = \Theta(S n!) \approx S \sqrt{2\pi n},(n/e)^n.}
> > $$
> > This is clearly much larger than any observation space of constant size that does not grow with $n$. If we view the observation space as a fixed board $C$, then the ratio between the latent and observation spaces is on the order of $\sim S n!$, which exhibits factorial (super-exponential) growth in $n$.
> >
> > We also agree that our previous wording “exponentially larger” was not fully rigorous. In the revised version of the main text, we have therefore softened this to state that the latent space is *much larger* than the observation space. In addition, we now provide a concrete worked example of the latent-space complexity for Sudoku in Appendix A.3, which formalizes the above calculation.
> >
> >
> >
> > > **Q2:** The system is majorly motivated from selection mechanism, however the system is more inherited from DEQ or Implicit Deep Learning frameworks. How does SR² differ theoretically from them? Is there any convergence guarantee for your fixed-point iterations?
> >
> >
> > Thank you for your insightful question, and we would like to take this chance to highlight the difference between this submission and DEQ. In brief, our method can be viewed as a truncated, explicitly unrolled variant of DEQ.
> >
> > The DEQ framework conceptually mimics an infinite-depth neural network by seeking an equilibrium point that represents the steady state of the transformation dynamics. In principle, this equilibrium corresponds to the limit of an infinite number of iterations. In practice, DEQ computes the gradient of the equilibrium state with respect to the parameters via the implicit function theorem, i.e., using a single gradient step without storing intermediate states.
> >
> > In contrast, our work explicitly unrolls the iterative process and updates the model using a finite number of refinement steps. Thus, rather than jumping directly to an equilibrium via an implicit gradient, we model and train the intermediate refinement trajectory itself.
> >
> > Importantly, this choice of finite, explicit updates is not only motivated by our causally inspired hypothesis, but also supported by empirical evidence. Specifically, we implemented a DEQ-style formulation and compared it with SR². In our experiments, directly computing a fixed point for DEQ required a very large number of iterations to obtain a sufficiently accurate approximation; moreover, its training performance was substantially worse than that of a simpler iterative scheme with repeated application of supervision signals at multiple steps. The results are summarized below and show that SR² achieves significantly better performance than the DEQ approach:
> >
> >
> > | Method | Params | Sudoku-Extreme |
> > |--------|--------|----------------|
> > | DEQ    | 27.3M  | 2.70           |
> > | $\text{SR}^{2}$    | 3.4M   | 66.63          |
> >
> >
> > Motivated by your question, we have added a dedicated comparison between DEQ and our framework in Section 2.3 of the revised submission.
> >
> > Moreover, regarding your concern about convergence guarantees, we now provide a proof sketch of the convergence properties of SR² in Appendix A.4 of the revised paper.
> >
> > ---
> > References
> >
> > [1] Zhu, R.-J., Wang, Z., Hua, K., Zhang, T., Li, Z., et al. (2025, October 29). *Scaling latent reasoning via looped language models* (arXiv:2510.25741). arXiv
> >
> > [2] Wah, C., Branson, S., Welinder, P., Perona, P., & Belongie, S. (2011). The Caltech-UCSD Birds-200-2011 dataset (CNS-TR-2011-001). California Institute of Technology.

---

> > > ### Comment · Reviewer_qWVm · 2025-11-26
> > > **Thanks for the response**
> > >
> > > Thanks for the detailed response. My score already represents the higher end. I decide to keep my original evaluation.

---

> > > > ### Author Response · Authors · 2025-11-26
> > > > **Thanks for your recognition**
> > > >
> > > > Dear Reviewer qWVm,
> > > >
> > > > Thank you for your positive feedback. We sincerely appreciate your recognition of our work.
> > > >
> > > > Best regards,
> > > >
> > > > Authors

---

### Author Response · Authors · 2025-12-01
**General Response Official Comment by Authors**

Dear Reviewers and AC,

We thank the reviewers and the area chairs for their careful evaluation of our work and for the constructive feedback provided. We are particularly grateful to Reviewer nGiF and Reviewer qWVm, who on November 26 kindly provided follow-up responses indicating their intention to raise a score from 4 to 6 and to maintain a positive score of 8. We also sincerely appreciate the efforts of the re-assigned area chair for devoting valuable time to read our manuscript and subsequent discussion. For clarity and convenience, we first summarize below the key strengths highlighted by the reviewers, followed by our corresponding revisions and responses to their concerns.

(*We refer to Reviewer qWVm as R1, Reviewer o7wn as R2, Reviewer v12F as R3, and Reviewer nGiF as R4)

**Key strengths noted by the reviewers:**

* Novel causal and latent-variable perspective on reasoning (R1, R2, R3)

* Coherent SR² architecture and reflection formulation (R1, R2, R4)

* Strong empirical performance and parameter efficiency (R1, R2, R3, R4)

**Common concerns and our responses:**

* **Additional STEM and ARC-AGI benchmarks (R2, R3, R4)**
  In response to the request for additional benchmarks, we have added experiments on the ARC-AGI benchmark (including ARC-1 and ARC-2) and now report comparative results for SR² and the baselines in $\color{green}{\text{Table 1}}$ of the revised manuscript.

* **Training and test cost analysis (R2, R4)**
  To address concerns about training and inference cost, we have added a dedicated efficiency analysis in $\color{green}{\text{Section 3.4}}$, comparing SR² and the baselines in terms of training speed, inference speed, and GPU memory usage.

**Other revisions addressing specific comments:**

* For R1 (W1), we pointed out the potential feasibility of scaling SR² to large-scale settings.

* For R1 (W2), we provided additional justification for the necessity of iterative refinement and alignment in reasoning tasks and added supporting experiments in $\color{green}{\text{Appendix E}}$.

* For R1 (Q1), we clarified in $\color{green}{\text{Section 2.2}}$ our statement about the relative sizes of latent and observed spaces in reasoning tasks and added a concrete example of Sudoku latent-space complexity in $\color{green}{\text{Appendix A.3}}$.

* For R1 (Q2), we included a detailed comparison between DEQ and SR² in $\color{green}{\text{Section 2.3}}$ and added an outline of SR²’s convergence properties in $\color{green}{\text{Appendix A.4}}$.

* For R2 (W1/Q1), we expanded $\color{green}{\text{Section 2.1}}$ with a more detailed explanation of Eq. (1).

* For R2 (Q3), we further clarified in $\color{green}{\text{Section 2.3}}$ the connection between the SR² theory and its concrete architecture.

* For R3 (W2), we added a discussion of LLM self-correction techniques to $\color{green}{\text{Appendix A}}$.

* For R3 (Q1), we further elaborated on the potential applications of SR² in LLM scaling.

* For R4 (Q1), we added an explicit example in $\color{green}{\text{Section 2.1}}$ to illustrate the concrete meanings of $X$, $Y$, $z$, and $S$.

* For R4 (Q2), we revised the discussion of Eq.(3) in $\color{green}{\text{Section 2.3}}$ to clarify the motivation for formalizing $z$ as a fixed-point equation.

Finally, we would like to once again express our sincere gratitude to the re-assigned AC for their time and effort. We understand that, due to an OpenReview system issue, re-assigned ACs had to review a large number of submissions within a very limited time window. We are deeply appreciative that, despite this heavy workload, you carefully read our revised manuscript and the full discussion process.

---

### Meta-Review · Area_Chair_CCmG · 2025-12-24

**Summary:**

The paper examines reasoning tasks from a causal standpoint. It develops a new framework inspired by causal analysis, which achieves promising results across 3 benchmarks.

All the reviewers found the proposed framework quite interesting. They felt that the empirical results were strong.

The main concern raised was a lack of evaluation across more benchmarks. In response, the authors have added evaluation on the ARC benchmark, which are also positive.

The other concern is regarding the writing, which I also agree with. The paper could be clearer in describing its framework. I suggest using explicit examples such as Sudoko to explain what the framework does in Section 2.3 and Fig. 3.

Due to the positive scores and the promise of the framework, I recommend that the paper be accepted. I strongly encourage the authors to make the paper more approachable and understandable, which will also help increase its impact and visibility.

**Reviewer Concerns:**

As discussed above, the main concern was regarding limited evaluation, and this has been addressed to a reasonable extent with the ARC results.

The other concern is regarding writing, where more improvements are needed.

**Reviewer Scores:**

It seems that there could have been multiple increases in scores for this paper. Reviewer nGiF did say they will increase their score. Reviewer o7wn wanted to see results on ARC, so they could also have increased their score from 4.

---

### Decision · Program_Chairs · 2026-01-26

Accept (Poster)